# Equivariant Graph Network Approximations of High-Degree Polynomials for Force Field Prediction

**Zhao Xu**[*]                                                                    *zhaoxu@tamu.edu*
*Department of Computer Science & Engineering*
*Texas A&M University*

**Haiyang Yu**[*]                                                                  *haiyang@tamu.edu*
*Department of Computer Science & Engineering*
*Texas A&M University*

**Montgomery Bohde**                                                        *mbohde2015@tamu.edu*
*Department of Computer Science & Engineering*
*Texas A&M University*

**Shuiwang Ji**                                                                        *sji@tamu.edu*
*Department of Computer Science & Engineering*
*Texas A&M University*

**Reviewed on OpenReview:** *https://openreview.net/forum?id=7DAFwpOVne*

## Abstract

Recent advancements in equivariant deep models have shown promise in accurately predicting atomic potentials and force fields in molecular dynamics simulations. Using spherical harmonics (SH) and tensor products (TP), these equivariant networks gain enhanced physical understanding, like symmetries and many-body interactions. Beyond encoding physical insights, SH and TP are also crucial to represent equivariant polynomial functions. In this work, we analyze the equivariant polynomial functions for the equivariant architecture, and introduce a novel equivariant network, named PACE. The proposed PACE utilizes edge booster and the Atomic Cluster Expansion (ACE) technique to approximate a greater number of $SE(3) \times S_n$ equivariant polynomial functions with enhanced degrees. As experimented in commonly used benchmarks, PACE demonstrates state-of-the-art performance in predicting atomic energy and force fields, with robust generalization capability across various geometric distributions under molecular dynamics (MD) across different temperature conditions. Our code is publicly available as part of the AIRS library `https://github.com/divelab/AIRS/`.

## 1 Introduction

Deep learning has led to notable progress in computational quantum chemistry tasks, such as predicting atomic potentials and force fields in molecular systems (Zhang et al., 2023). Fast and accurate prediction of energy and force is desired, as it plays crucial roles in advanced applications such as material design and drug discovery. However, it is insufficient to rely solely on learning from data, as there are physics challenges that must be taken into consideration. For example, to better consider symmetries inherent in 3D molecular structures, equivariant graph neural networks (GNNs) have been developed in recent years. By using equivariant features and equivariant operations, SE(3)-equivariant GNNs ensure equivariance of permutation, translation and rotation. Thus, their internal features and predictions transform accordingly as the molecule is rotated or translated. Existing equivariant GNNs can specialize in handling features with either rotation order $\ell = 1$ (Schütt et al., 2021; Jing et al., 2021; Satorras et al., 2021; Du et al., 2022; 2023;

---

[*]Equal contribution.

Thölke & Fabritiis, 2022) or higher rotation order $\ell > 1$ (Thomas et al., 2018; Fuchs et al., 2020; Liao & Smidt, 2023; Batzner et al., 2022; Batatia et al., 2022a;b; Unke et al., 2021a; Yu et al., 2023b; Li et al., 2022), while invariant methods only consider rotation order $\ell = 0$ (Schütt et al., 2017; Smith et al., 2017; Chmiela et al., 2017; Zhang et al., 2018a;b; Schütt et al., 2018; Ying et al., 2021; Luo et al., 2023; Gasteiger et al., 2020; Liu et al., 2022; Gasteiger et al., 2021). Generally, methods with higher rotation order exhibit improved performance but at the cost of higher computational complexity. In addition to rotation order, some equivariant methods (Musaelian et al., 2023; Batatia et al., 2022a;b) also consider many-body interactions in their model design. These approaches follow traditional principles (Brown et al., 2004; Braams & Bowman, 2009) of decomposing the potential energy surface (PES) as a linear combination of body-ordered functions. In contrast to standard message passing (Gilmer et al., 2017) that considers interactions between two atoms in each message, many-body methods aim to incorporate the interactions of multiple atoms surrounding the central node.

Interestingly, equivariant neural networks leveraging spherical harmonics as edge features can be interpreted from another perspective of approximating equivariant polynomial functions. This stems from the ability of spherical harmonics to represent both invariant and equivariant polynomial functions via tensor products. Additionally, the tensor contraction operation introduced by Atomic Cluster Expansion (ACE) (Drautz, 2019; Dusson et al., 2022; Kovács et al., 2021) broaden the model's scope, covering a wider range of polynomial functions. Thus, the primary aim of this work is to develop a model that exhibits enhanced expressiveness, particularly for predicting atomic energy and force fields.

In the present study, we expand the scope of equivariant polynomial function analysis, transitioning from point cloud networks to equivariant networks that aim to predict symmetric physical properties, such as atomic energy. This analytical extension informs the development of our novel equivariant network, PACE. Our proposed network effectively integrates an edge booster and many-body interactions through the ACE technique, which demonstrates an advanced capacity for approximating $SE(3) \times S_n$ equivariant functions, with higher polynomial degrees. Our method is termed PACE as it is based on polynomial function approximation and ACE. To evaluate the efficacy of our approach, we assess our model on three molecular dynamics simulation datasets, including rMD17, 3BPA and AcAc, and obtain consistent performance enhancements. Notably, our model not only achieves state-of-the-art performance in energy and force predictions but also exhibits strong generalization capabilities in geometric out-of-distribution settings under MD across different temperatures.

## 2 Symmetries and Polynomial Functions

### 2.1 Symmetries

Incorporating physical symmetries into machine learning models is pivotal for tackling quantum chemistry challenges. This is because numerous quantum properties of molecules inherently exhibit equivariance or invariance to symmetry transformations. For instance, if we rotate a molecule in 3D space, forces acting on atoms rotate accordingly while the total energy of the molecule remains invariant. Besides rotation, the permutation of atoms within a molecule represents another type of symmetry. In such scenarios, the forces on atoms obey permutation equivariance, while the molecular energy remains invariant to atomic permutations. To summarize, atomic forces can be characterized as $SE(3)$-equivariant to rotational transformations and $S_n$-equivariant to atomic permutations, reflecting the respective symmetry groups $G$ associated with these symmetry operations. Concurrently, molecular energy exhibits $SE(3)$-invariance and $S_n$-invariance.

Mathematically (Bronstein et al., 2021), given a group $G$ and group action $*$, we say a function $f$ mapping from source domain $Q$ to target domain $Y$ is $G$-equivariant if $f(g * \mathbf{q}) = g * f(\mathbf{q}), \forall \mathbf{q} \in Q, \forall g \in G$. Similarly, if $f(g * \mathbf{q}) = f(\mathbf{q})$ holds, we say $f$ is $G$-invariant. Due to the intrinsic $SE(3)$ and $S_n$ equivariant and invariant symmetries in quantum chemistry, it is natural to encode these symmetries directly into the model architectures for effectively approximating the function $f$.

## 2.2 Polynomial Functions in Equivariant GNNs

Building on this concept of learning a powerful invariant or equivariant function $f$, various equivariant networks have been developed. These networks (Thomas et al., 2018; Batatia et al., 2022b; Yu et al., 2023b) are designed for mapping 3D molecular structures to inherently symmetric properties, such as energy, force, and the Hamiltonian matrix. Typically, these networks capture directions $\hat{r}$ and distances $\bar{r}$ by using real spherical harmonics $Y(\hat{r})$ combined with radial basis functions (rbf). Owing to their equivariance property, spherical harmonics are advantageous for encoding geometric information and predicting the physical properties of molecules.

It is noteworthy that spherical harmonics, as partially discussed in Dusson et al. (2022), can also serve as proper features for representing $SE(3)$ invariant and equivariant polynomial functions with the help of tensor products. Here, we illustrate how spherical harmonics represent polynomial functions using a specific example. We start with a polynomial function without $SE(3)$ invariance and equivariance constrain. In this polynomial function denoted as $P^D(x, y, z) : \mathbb{R}^3 \to \mathbb{R}^1$, $D$ is the highest degree of the polynomial terms, and $\hat{r} = (x, y, z)$. By observing Equations (1) to (3), we can find polynomial terms of $P^D(x, y, z)$ with $D \leq 2$ in $Y^{\ell=0}, Y^{\ell=1}$, and $Y^{\ell=1} \otimes Y^{\ell=1}$, where $\otimes$ denotes tensor product, $\mathbf{C}$ is the coefficients of the spherical harmonics, and $Y^{\ell=1}(\hat{r})^{\otimes t}$ represents the tensor product of $Y^{\ell=1}(\hat{r})$ with itself, repeated $t$ times. That's to say, the polynomial functions $P^{D \leq 2}(x, y, z)$ can be represented as a linear combination of entries in $Y^{\ell=0}, Y^{\ell=1}$, and $Y^{\ell=1} \otimes Y^{\ell=1}$.

Next, let's consider the $SE(3)$ equivariant polynomial function $P^{D \leq 2}_{SE(3)}(x, y, z) : \mathbb{R}^3 \to \mathbb{R}^K$, and we use $P^{D \leq 2}_{SE(3)}[k]$ to denote the $k$-th output of this function, where $1 \leq k \leq K$. Note that each output of this function is still a polynomial function, which is denoted as $P^{D \leq 2}_{SE(3)}[k] \in \{P^D(x, y, z) | D \leq 2\}$. Therefore, there exists a linear function mapping from $Y^{\ell=0}(\hat{r}), Y^{\ell=1}(\hat{r}), Y^{\ell=1}(\hat{r})^{\otimes 2}$ into $P^{D \leq 2}_{SE(3)}[k](x, y, z), 1 \leq k \leq K$, which then comprise $P^{D \leq 2}_{SE(3)}(x, y, z)$. Given that both the input features and polynomial functions are $SE(3)$ equivariant, the corresponding linear mapping also retains this equivariance. From this, we can conclude that for any polynomial function $P^{D \leq 2}_{SE(3)}(x, y, z)$, there exists an equivariant linear function mapping from $Y^{\ell=0}(\hat{r}), Y^{\ell=1}(\hat{r})$, and $Y^{\ell=1}(\hat{r})^{\otimes 2}$ into this function. This principle of linear universality is also discussed in Dym & Maron (2020).

$$Y^{\ell=0}(\hat{r}) = \mathbf{C}_{\ell=0} \cdot \begin{bmatrix} 1 \end{bmatrix} \tag{1}$$

$$Y^{\ell=1}(\hat{r}) = \mathbf{C}_{l=1} \cdot \begin{bmatrix} y \\ z \\ x \end{bmatrix} \tag{2}$$

$$Y^{\ell=1}(\hat{r}) \otimes Y^{\ell=1}(\hat{r}) = Y^{\ell=1}(\hat{r})^{\otimes 2} = \mathbf{C}_{l=1}\mathbf{C}_{l=1}^T \cdot \begin{bmatrix} y^2 & yz & yx \\ zy & z^2 & zx \\ xy & xz & x^2 \end{bmatrix} \tag{3}$$

As previously elucidated, spherical harmonics and their tensor products provide a way to approximate the polynomial functions, ensuring the preservation of $SE(3)$-equivariance. In existing equivariant graph networks (Batzner et al., 2022; Batatia et al., 2022b; Liao & Smidt, 2023), spherical harmonics usually serve as components of edge messages to provide $SE(3)$ equivariant features. For $S_n$ permutation equivariance, Graph Neural Networks (GNNs)(Kipf & Welling, 2017; Veličković et al., 2018; Gao & Ji, 2019; Liu et al., 2020; Cai et al., 2021; Chen et al., 2024) employ a message passing scheme to aggregate neighboring messages for each central node. Therefore, by adhering to the aggregation of this message passing scheme, $SE(3)$-equivariant features can successfully attain $S_n$ equivariance. Specifically, edge messages are constructed using spherical harmonics $Y^{\ell=1}(\hat{r}_{ij})$, leading to aggregated equivariant features that are analogous to

$$f_{\mathbf{x}_i} = \sum_{j \in \mathcal{N}_i} Y^{\ell=1}(\hat{r}_{ij}). \tag{4}$$

Equivariant GNNs like Anderson et al. (2019) and Batatia et al. (2022b) further implement architectures akin to

$$f'_{\mathbf{x}_i} = \sum_{j_1 \in \mathcal{N}_i} Y^{\ell=1}(\hat{r}_{ij_1}) \otimes \sum_{j_2 \in \mathcal{N}_i} Y^{\ell=1}(\hat{r}_{ij_2}), \tag{5}$$

using tensor product within their update modules. Both the basic aggregated message $f_{\mathbf{x}_i}$ and its updated form $f'_{\mathbf{x}_i}$ qualify as $SE(3) \times S_n$ equivariant polynomial functions with the highest degree $D \leq 2$. However, they do not cover all $SE(3) \times S_n$ equivariant polynomial functions with the highest degree $D$. For example,

$$\hat{f}_{\mathbf{x}_i} = \sum_{j \in \mathcal{N}_i} Y^{\ell=1}(\hat{r}_{ij})^{\otimes 2}, \tag{6}$$

is also a $SE(3) \times S_n$ equivariant polynomial function with $D = 2$, but it is not covered by $f'_{\mathbf{x}_i}$ as demonstrated in appendix E. Therefore, developing a model architecture capable of covering a broader range of equivariant polynomial functions is crucial for enhancing its expressiveness.

### 2.3 Atomic Energy with Local Environment for Equivariant Polynomial Functions

The approximation of polynomial functions in point cloud networks has been explored previously, as detailed in Section 3.1 (Maron et al., 2019; Keriven & Peyré, 2019; Dym & Maron, 2020). In this series of analyses, the architecture's capacity to approximate invariant or equivariant polynomial functions is demonstrated, using $3D$ coordinates $(\mathbf{p}_1, \mathbf{p}_2, \cdots, \mathbf{p}_n)$ as inputs. However, the analysis for equivariant networks in predicting symmetric physical properties is still underexplored. In energy and force prediction tasks, atomic cluster expansion (ACE) is a widely used technique for approximating atomic energy. ACE decomposes the total energy $E_i$ into a sum of individual atomic energies $E_i$ for each atom $i$, and then uses local environments to learn these atomic energies, formulated as

$$
\begin{aligned}
E_i(\boldsymbol{\theta}_i) = &\sum_j \sum_v c_v^{(1)} \phi_v(\mathbf{r}_{ij}) + \\
&\frac{1}{2} \sum_{j_1 j_2} \sum_{v_1 v_2} c_{v_1 v_2}^{(2)} \phi_{v_1}(\mathbf{r}_{ij_1}) \phi_{v_2}(\mathbf{r}_{ij_2}) + \\
&\frac{1}{3!} \sum_{j_1 j_2 j_3} \sum_{v_1 v_2 v_3} c_{v_1 v_2 v_3}^{(3)} \phi_{v_1}(\mathbf{r}_{ij_1}) \phi_{v_2}(\mathbf{r}_{ij_2}) \phi_{v_3}(\mathbf{r}_{ij_3}) \\
&+ \cdots,
\end{aligned}
\tag{7}
$$

where $\boldsymbol{\theta}_i = (\mathbf{r}_{ij_1}, \cdots, \mathbf{r}_{ij_N})$ denotes $N$ edges in the atomic environment, $\mathbf{r}_{ij}$ denotes edge direction and distance from atom $i$ to atom $j$, $\phi$ denotes the single edge basis function, and $c$ represents the coefficients. Therefore, in the context of approximating atomic energy, as opposed to their use in point cloud networks, the polynomial function of node features $f_{\mathbf{x}_i}$ here focuses on the $N$-edge system $\boldsymbol{\theta}_i$ rather than on atomic coordinates.

The research by Dym & Maron (2020) introduces the concept of $D$-spanning function sets to denote the capability of covering $SE(3) \times S_n$ equivariant polynomial function with the highest degree $D$. Expanding upon their work, a $D$-spanning function for approximating atomic energy within an $N$-bond system is defined as

$$Q_K^{(\mathbf{t})}(\boldsymbol{\theta}_i) = \sum_{j_1, j_2, \ldots, j_K = 1}^{N} \mathbf{r}_{ij_1}^{\otimes t_1} \otimes \mathbf{r}_{ij_2}^{\otimes t_2} \otimes \ldots \otimes \mathbf{r}_{ij_K}^{\otimes t_K}, \tag{8}$$

where $K \geq D$, and $\mathbf{t} = (t_1, \cdots, t_K)$. Then, a $D$-spanning set is defined as

$$Q_K^D = \left\{ \iota \circ Q_K^{(\mathbf{t})}(\boldsymbol{\theta}_i) \mid \|\boldsymbol{t}\|_1 \leq D \right\}, \tag{9}$$

where $\iota$ denotes an equivariant linear function mapping. In Appendix A.5, we provide the mathematical definition of $D$-spanning, and elucidate the relationship between $D$-spanning function sets and the model's expressiveness.

## 2.4 Motivation of PACE

Inspired by the goal of approximating $SE(3) \times S_n$ equivariant functions with full spanning and higher polynomial degrees for the atomic energy based on the local atomic environment, in this work, we propose an equivariant network PACE following the message passing scheme. To briefly summarize using the same example from Equation (4), PACE incorporates a novel edge booster that leverages the tensor product of spherical harmonics from $\mathbf{r}_{ij}^{\otimes t}, t \leq \ell_{\max}$ to construct edge messages analogous to $\mathbf{r}_{ij}^{\otimes t}, t \leq \ell_{\max} * N_{\text{boost}}$. Hence, the aggregated equivariant features in PACE become

$$f_{\mathbf{x}_i} = \sum_{j \in \mathcal{N}_i} \mathbf{r}_{ij}^{\otimes t}, t \leq \ell_{\max} * N_{\text{boost}} \tag{10}$$

Then, PACE utilizes an update module with the symmetric contraction module from MACE to conduct many-body interaction with correlation $v$ and the corresponding equivariant polynomial function defined as

$$\hat{f}_{\mathbf{x}_i} = \sum_{j_1, \cdots, j_v \in \mathcal{N}_i} \underbrace{\mathbf{r}_{ij_1}^{\otimes t_1} \otimes \cdots \otimes \mathbf{r}_{ij_{v_{\max}}}^{\otimes t_{v_{\max}}}}_{v_{\max} \text{ times}} t_1, \cdots, t_v \leq \ell_{\max} * N_{\text{boost}}. \tag{11}$$

Consequently, the updated equivariant features $\hat{f}_{\mathbf{x}_i}$ can effectively cover $SE(3) \times S_n$ equivariant functions with formulation Equation (11) and achieve $D$-spanning with $D = \min\{\ell_{\max} * N_{\text{boost}}, v_{\max}\}$.

# 3 Related Works

## 3.1 Universality Analysis

Universality is a powerful property for neural networks that can approximate arbitrary functions. While Zaheer et al. (2017); Maron et al. (2019); Keriven & Peyré (2019) study the universality of permutation invariant networks, several works have recently studied the rotational equivariant networks. Dym & Maron (2020) takes use of the proposed tensor representation to build $D$-spanning family and shows that Tensor Field Networks (TFN) (Thomas et al., 2018) is proved to be a universal equivariant network capable of approximating arbitrary equivariant functions defined on the point coordinates of point cloud data. Furthermore, GemNet (Gasteiger et al., 2021) uses the conclusion in Dym & Maron (2020), and is proved to be a universal GNN with directed edge embeddings and two-hop message passing.

## 3.2 Equivariant Graph Neural Networks

In recent years, equivariant graph neural networks have been developed for 3D molecular representation learning, as they are capable of effectively incorporating the symmetries required by the specific task (Ruddigkeit et al., 2012; Chmiela et al., 2017; Yu et al., 2023a; Khrabrov et al., 2022). Existing equivariant 3D GNNs can be broadly classified into two categories, depending on whether they utilize order $\ell = 1$ equivariant features or higher order $\ell > 1$ equivariant features. Methods belonging to the first category (Satorras et al., 2021; Schütt et al., 2021; Deng et al., 2021; Jing et al., 2021; Thölke & Fabritiis, 2022) achieve equivariance by applying constrained operations on order 1 vectors, such as vector scaling, summation, linear transformation, vector product, and scalar product. The second category of methods (Thomas et al., 2018; Fuchs et al., 2020; Liao & Smidt, 2023; Batzner et al., 2022; Batatia et al., 2022a;b; Brandstetter et al., 2021) predominantly employs tensor products (TP) to preserve higher-order equivariant features, with some works (Luo et al., 2024) focusing on accelerating tensor product computations.

## 3.3 Atomic Cluster Expansion

Molecular potential and force field are crucial physical properties in molecular analysis. To approximate these properties, the atomic cluster expansion (ACE) (Drautz, 2019; Kovács et al., 2021) is used to approximate the atomic potential. Recently, several neural networks aiming to predict atomic potential and forces have been developed to consider many-body interactions by incorporating ACE into their model architectures. Specifically, BOTNet (Batatia et al., 2022a) takes multiple message passing layers to encode the many-body

interaction and analyzes the body order for various message passing schemes. MACE (Batatia et al., 2022b) takes generalized Clebsch-Golden coefficients to couple the aggregated message to incorporate higher-order interactions. Allegro (Musaelian et al., 2023) uses a series of tensor product layers to calculate the equivariant representations without using message passing, learning many-body interactions.

### 3.4 Comparison to Existing Works

A NequIP (Batzner et al., 2022) layer builds its message using the original spherical harmonics, and its output irreducible representations can cover equivariant polynomials defined as $\sum_{j \in \mathcal{N}_i} \mathbf{r}_{ij}^{\otimes t}, t \leq \ell_{\max}$, which can form a $D$-spanning family with the highest degree $D = 1$. For a MACE layer, the output irreducible representations can cover the equivariant polynomials defined as $Q_{K=v_{\max}}^{(\mathbf{t})} = \sum_{j_1,\cdots,j_{v_{\max}} \in \mathcal{N}_i} \underbrace{\mathbf{r}_{ij_1}^{\otimes t_1} \otimes \cdots \otimes \mathbf{r}_{ij_{v_{\max}}}^{\otimes t_{v_{\max}}}}_{v_{\max} \text{ times}}, t_1, \cdots, t_{v_{\max}} \leq \ell_{\max}.$ , which can form a $D$-spanning family with degree $D = \max\{\ell_{\max}, v_{\max}\}$. For the proposed PACE layer, the output irreducible representations cover $Q_{K=v_{\max}}^{(\mathbf{t})} = \sum_{j_1,\cdots,j_{v_{\max}} \in \mathcal{N}_i} \underbrace{\mathbf{r}_{ij_1}^{\otimes t_1} \otimes \cdots \otimes \mathbf{r}_{ij_{v_{\max}}}^{\otimes t_{v_{\max}}}}_{v_{\max} \text{ times}}, t_1, \cdots, t_{v_{\max}} \leq \ell_{\max} * N_{\text{boost}}$, spanning a $D$-spanning family with $D = \max\{\ell_{\max} * N_{\text{boost}}, v_{\max}\}$, as explained in Section 2.4. During our experiments, we set $\ell_{\max} = 3$ and $v_{\max} = 3$ following previous work (Batatia et al., 2022b; Musaelian et al., 2023) for a fair comparsion. Although both MACE and PACE can form a $D$-spanning family with $D = 3$, our PACE model can still approximate a greater number of polynomial functions with $D > 3$ in this scenario. Moreover, by using additional self-interactions, PACE needs fewer channels to approximate the same amount of positions in D-spanning functions.

## 4 The Proposed PACE

The node features $\mathbf{x}_i^0$ are initialized through a linear transformation applied to its atomic type. Edges constructed based on cutoff distance have orientations $\hat{r}_{ij}$ denoted by spherical harmonics $Y^\ell(\hat{r}_{ij})$, and pairwise distances $\bar{r}_{ij}$ embedded using a learnable radial basis function $R$. The proposed PACE comprises two distinct message passing layers followed by an output layer. The illustration of PACE architecture is provided in Figure 1. In this section, we will introduce the architecture of our proposed PACE model as well as the corresponding equivariant polynomial functions from the perspective of irreducible representations. Detailed information about the irreducible representations used in equivariant GNNs can be found in Appendix A.1. As shown in Appendix B.1, the irreducible representations can also represent a $D$-spanning family.

**Theorem 4.1.** *For any $D$-spanning function $Q_K^{(\mathbf{t})}(\boldsymbol{\theta}_i)$ appeared in $Q_K^D$ and for any position $P = (p_1, p_2, \cdots, p_k)$ in tensor representation, where $p_k \in \mathbb{R}^3$ denotes the element position, if there exists $w_1$ and irreps$_1$ such that $Q_K^{(\mathbf{t})}(\boldsymbol{\theta}_i)(P) = \sum_{\ell m} w_1^{\ell m} irreps_1^{\ell m}$, and the set of irreps$_1$ forms a $D$-spanning family.*

**Definition 4.2.** *If the set of irreps$_1$ forms a D-spanning family $Q_K^{(\mathbf{t})}(\boldsymbol{\theta}_i)$, we can say that the irreducible representation irreps$_1$ represents $Q_K^{(\mathbf{t})}(\boldsymbol{\theta}_i)$.*

### 4.1 The First Layer

Following the message passing neural network, the first step is to build edge messages from neighboring nodes to the central node. As discussed in Section 2.4, the construction of these edge messages for node pairs in equivariant neural networks generally requires a tensor product to combine edge spherical harmonics and node features at each layer. Diverging from this conventional design, the first layer of PACE uses the edge booster two consecutive tensor products, which means $N_{\text{boost}} = 2$, to construct the edge message as

$$\mathbf{m}_{ij,1}^1 = Y(\hat{r}_{ij}) \otimes_{w_{1,ij}} \text{MLP}(\mathbf{x}_i^0 \parallel \mathbf{x}_j^0), \tag{12}$$

$$\mathbf{m}_{ij,2}^1 = Y(\hat{r}_{ij}) \otimes_{w_{2,ij}} \mathbf{m}_{ij,1}^1, \tag{13}$$

where the learnable weights $w_{1,ij}$ and $w_{2,ij}$ applied to each tensor product are obtained by

$$w_{1,ij} = \text{MLP}(R(\bar{r}_{ij})), \tag{14}$$

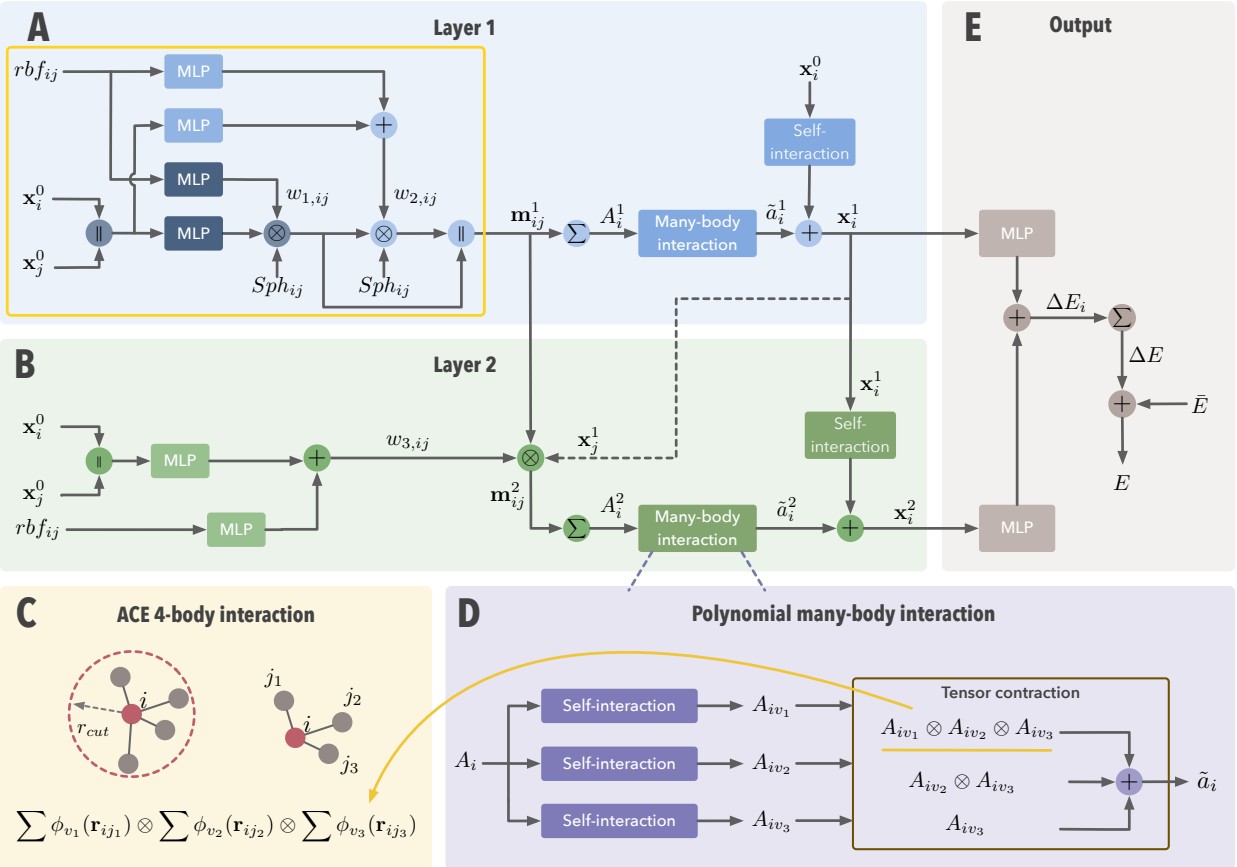

Figure 1: An architecture overview of PACE. **A:** The first message passing layer. The initial node features $\mathbf{x}_i^0$ and $\mathbf{x}_j^0$ are concatenated and transformed by an MLP. Then, a tensor product is applied to the transformed node features and edge spherical harmonics $Sph_{ij}$ with an RBF and MLP transformed edge distance as learnable weights. Next, its output is further combined with $Sph_{ij}$ using another tensor product, and the weights involve the initial node features and edge distance. These operations in the yellow box comprise the edge booster, aiming to enhance the model's expressiveness. The obtained edge-boosted messages are aggregated to form the atomic base. Then, an update module comprising a polynomial many-body interaction module and a skip connection is used to update the features of central nodes. **B:** The second message passing layer. A tensor product is applied to the edge message and updated node features that are obtained from the first layer. The learnable weights of the tensor product are based on the initial node features and distance of each neighboring pair. Then, edge messages are aggregated and node features are updated similarly to those in the first layer. **C:** An example of 4-body interaction in ACE. We aim to fit $\sum \phi_{v_1}(\mathbf{r}_{ij_1}) \otimes \sum \phi_{v_2}(\mathbf{r}_{ij_2}) \otimes \sum \phi_{v_3}(\mathbf{r}_{ij_3})$ using $A_{iv_1} \otimes A_{iv_2} \otimes A_{iv_3}$, where $\{\phi\}$ denotes the atomic base in ACE. **D:** Polynomial many-body interaction module. The atomic base $A_i$ is fed to multiple self-interaction layers separately to produce different $A_{iv}$. Then, tensor contraction is performed to produce $\tilde{a}_i$. **E:** Output. The invariant part of node features produced by both layers are transformed and summed to predict the deviation from the total molecular energy to its average.

$$w_{2,ij} = \text{MLP}(R(\bar{r}_{ij})) + \text{MLP}(\mathbf{x}_i^0 \parallel \mathbf{x}_j^0), \tag{15}$$

considering the atomic types for node pairs and their pairwise distances. Then, we obtain the edge message by concatenating these two together, denoted as

$$\mathbf{m}_{ij}^1 = (\mathbf{m}_{ij,1}^1 \parallel \mathbf{m}_{ij,2}^1). \tag{16}$$

As shown in Appendix B.1, the edges features from spherical harmonics represent $\mathbf{r}_{ij}^{\otimes t}, t \leq \ell_{\max}$, with $\ell_{\max} = 3$. Then the second tensor product increases it to $\mathbf{r}_{ij}^{\otimes t}, t \leq N_{\text{boost}} * \ell_{\max}$ with $N_{\text{boost}} = 2$ using Lemma B.2.

After obtaining edge messages via the edge booster, we aggregate neighboring messages to the central node as

$$A_i^1 = \frac{1}{|\mathcal{N}(i)|} \sum_{j \in \mathcal{N}(i)} \mathbf{m}_{ij}^1, \tag{17}$$

where $|\mathcal{N}(i)|$ is the number of neighbor nodes. Here, $A_i$ is analogous to the atomic base of the central node in the original ACE method (Drautz, 2019; Dusson et al., 2022; Kovács et al., 2021). Meanwhile, the equivariant polynomial form of $A_i$ is $\sum_{j \in \mathcal{N}_i} \mathbf{r}_{ij}^{\otimes t}, t \leq 6$.

Next, $A_i$ is processed by a polynomial many-body interaction module to produce features $\tilde{a}_i$. The corresponding equivariant polynomial form of $\tilde{a}_i$ is $Q_{K=3}^{(\mathbf{t})}(\boldsymbol{\theta}_i) = \sum_{j_1,j_2,j_3 \in \mathcal{N}_i} \mathbf{r}_{ij_1}^{\otimes t_1} \otimes \mathbf{r}_{ij_2}^{\otimes t_2} \otimes \mathbf{r}_{ij_3}^{\otimes t_3}, t_1, t_2, t_3 \leq 6$ for $P^D(\boldsymbol{\theta}_i)$ with $D \leq 3$ as shown in Appendix B.2.2. These features are then combined with the initial node features to update for each node. Details of the polynomial many-body interaction module are described in Section 4.3, and the design of the first layer is illustrated in Figure 1A.

## 4.2 The Second Layer

The second layer of PACE (Figure 1B) involves only one tensor product to construct the message. However, unlike NeuquIP and MACE, which leverage the original edge spherical harmonics, we instead use the edge-boosted message $\mathbf{m}_{ij}^1$ obtained from the first layer. Specifically,

$$\mathbf{m}_{ij}^2 = \mathbf{m}_{ij}^1 \otimes_{w_{3,ij}} \mathbf{x}_j^1, \tag{18}$$

where $\mathbf{x}_i^1$ denotes the updated node features from the first layer and the learnable weights are

$$w_{3,ij} = \text{MLP}(R(\bar{r}_{ij})) + \text{MLP}(\mathbf{x}_i^0 \| \mathbf{x}_j^0). \tag{19}$$

Next, messages are aggregated for the central nodes. Finally, another polynomial many-body interaction module is applied to update node features. The $L = 0$ invariant features outputted by both layers are further used for energy prediction.

## 4.3 Polynomial Many-Body Interaction Module

The polynomial many-body interaction module playing an important role in both PACE layers is proposed to incorporate many-body interactions by mixing the atomic base $A_i$. As shown in Figure 1D, we first use $v_{\max}$ different self-interactions to map the input atomic base $A_i$ to different bases $A_{iv}$ following

$$A_{ivc'}^\ell = \begin{cases} W_{vc'c}^\ell A_{ic}^\ell + b & \ell = 0 \\ W_{vc'c}^\ell A_{ic}^\ell & \ell > 0, \end{cases} \tag{20}$$

where $\ell$ is the rotation order of irreps, $c$ is the channel index for $A_i$ and $c'$ is the channel index for $A_i$. Then, we use tensor contraction (Batatia et al., 2022b) with generalized Clebsch-Golden to fuse multiple atomic bases. Tensor contraction is illustrated in Appendix A.4

As shown in Equation (25) in Appendix, in the case of two irreps, Clebsch-Gordan coefficients $C_{\ell_1 m_1, \ell_2 m_2}^{\ell_3 m_3}$ are used to maintain equivariance when fusing two irreps with rotation orders $\ell_1$ and $\ell_2$ to the output $\ell_3$, and the triplet $(\ell_1, \ell_2, \ell_3)$ is defined as a path. When fusing $N$ irreps, the generalized Clebsch-Gordan coefficients used to maintain the equivariance can be defined as

$$C_{\ell_1 m_1, \dots, \ell_n m_n}^{\mathcal{L}[N]\mathcal{M}[N]} = C_{\ell_1 m_1, \ell_2 m_2}^{L_2 M_2} C_{L_2 M_2, \ell_3 m_3}^{L_3 M_3} \cdots C_{L_{N-1} M_{N-1}, \ell_N m_N}^{L_N M_N}, \tag{21}$$

where $\mathcal{L}[N] = (\ell_1, L_2, \cdots L_N)$ with $|L_{i-1} - \ell_i| \leq L_i \leq |L_{i-1} + \ell_i|, L_i \in \mathbb{N}, \forall i \geq 2, i \in \mathbb{N}_+$, and the path is shown as $\eta[N] = (\ell_1, \ell_2, L_2, \ell_3, L_3, \cdots, \ell_{N-1}, L_{N-1}, \ell_N, L_N)$. Then, the output irreps is contracted one by

one to consider the coupled many-body interactions shown as

$$\tilde{a}_{i(v-1),L_{v-1}M_{v-1}}^{L_N M_N} = \sum_{\ell_v} \sum_{m_v=-\ell_v}^{\ell_v} A_{iv,l_v m_v} \sum_{\eta[v]} \left( W_{v,\eta} * \mathcal{C}_{\ell_1 m_1,\ldots,\ell_v m_v}^{\mathcal{L}[v]\mathcal{M}[v]} + \tilde{a}_{iv,L_v M_v}^{L_N M_N} \right), \tag{22}$$

where $W$ is the path weight, $\tilde{a}$ is the intermediate irreps, and $v \in \mathbb{N}_+$ starts from $N$ to 1 to incorporate N-body interactions. Note that $\mathcal{C} = 1$ and $L_0 = 0$ when $v = 1$, and $\tilde{a} = 0$ when $v = N$.

In contrast to MACE (Batatia et al., 2022b) applying tensor contraction to the atomic base $A_i$, our PACE first takes multiple self-interactions to distinguish the atomic base from the same $A_i$ to different $A_{iv}$ for different body orders before applying the contraction. According to Darby et al. (2023), the output of tensor contraction in MACE is defined as the tensor decomposed product basis while the one obtained in PACE is called tensor sketched product basis. Although Darby et al. (2023) has demonstrated that the tensor-decomposed basis can consistently reconstruct the tensor-sketched basis with a factor of $v_{\max}$ times channels, we implement the latter one in PACE to improve the model expressiveness without increasing the number of channels.

## 4.4 Output

As illustrated in Figure 1E, to compute the total energy of the molecule, we extract and transform the invariant part of node features produced by each PACE layer as

$$E = \bar{E} + \sum_{i=1}^{N} (\mathrm{MLP}(\mathbf{x}_{i,\ell m=00}^1) + \mathrm{MLP}(\mathbf{x}_{i,\ell m=00}^2)), \tag{23}$$

where $\bar{E}$ is the averaged total energy of the training set, and $N$ represents the number of atoms in a molecule. Once the total energy is predicted, we then use $\mathbf{f}_i = -\frac{\partial E}{\partial \mathbf{p}_i}$ to calculate the force acting on each atom, as it ensures energy conservation.

## 4.5 Summary of Key Contributions

**Edge Booster** The first key contribution of our work is the edge booster, which is computed by Equations (12) to (16) and illustrated as the yellow-boxed operations in Figure 1A. Our edge booster leverages consecutive tensor products on spherical harmonics which helps enhance an equivariant model's ability to approximate higher-degree polynomial functions. This idea is shown in Equation (10) and the overall polynomial degree is shown in Appendix B.2.2. There might be various implementations to build an edge booster following the idea of Equation (10). However, we choose to design our edge booster as described by Equations (12) to (16) for two reasons. First, our edge-boosted message only considers spherical harmonics and atomic types of nodes. Note that the computation of our edge booster does not involve node features that are updated layer by layer. Such a design follows our motivation to create an independent edge booster module that can produce an edge-boosted message to replace the original edge spherical harmonics used in each layer. Based on this design, we only need to compute the edge booster once in the first layer and we can reuse the edge-boosted message in all other layers. As shown in Figure 1A&B, the edge-boosted message $\mathbf{m}_{ij}^1$ is used in both layers. Hence, the design of our edge booster also considers computational costs. In summary, the result of our edge booster can be used in each layer to help approximate more polynomial functions with higher degree in an efficient manner. A detailed cost analysis of edge booster is provided in Appendix D.2.3.

**Additional Self-interaction Layers** The second enhancement comes from the additional self-interaction layers used in the polynomial many-body interaction module. In Section 4.3, we use the tensor contraction proposed in MACE (Batatia et al., 2022b) to approximate many-body interactions, but we apply additional self-interactions (SI) to the atomic base $A_i$ so that we use different inputs of the tensor contraction. In MACE, the atomic base $A_i$ is directly fed into the tensor contraction, while the atomic base $A_i$ in our design is first fed to multiple self-interaction layers separately to produce different atomic bases $A_{iv}$. The functionality of these self-interactions is to increase the ability to approximate more positions in $D$-spanning functions.

Table 1: Performance on the rMD17 dataset. Mean absolute errors (MAE) are reported for both energy (E) and force (F) predictions, with meV and meV/Å as units, respectively. Bold numbers highlight the best performance.

| | | ACE | FCHL | GAP | ANI | GemNet (T/Q) | NequIP | BOTNet | Allegro | MACE | Ours |
|---|---|---|---|---|---|---|---|---|---|---|---|
| Aspirin | E | 6.1 | 6.2 | 17.7 | 16.6 | - | 2.3 | 2.3 | 2.3 | 2.2 | **1.7** |
| | F | 17.9 | 20.9 | 44.9 | 40.6 | 9.5 | 8.2 | 8.3 | 7.3 | 6.6 | **5.8** |
| Azobenzene | E | 3.6 | 2.8 | 8.5 | 15.9 | - | 0.7 | 0.7 | 1.2 | 1.2 | **0.5** |
| | F | 10.9 | 10.8 | 24.5 | 35.4 | - | 2.9 | 3.3 | 2.6 | 3.0 | **2.2** |
| Benzene | E | 0.04 | 0.35 | 0.75 | 3.3 | - | 0.04 | 0.03 | 0.4 | 0.4 | **0.02** |
| | F | 0.5 | 2.6 | 6.0 | 10.0 | 0.5 | 0.3 | 0.3 | **0.2** | 0.3 | **0.2** |
| Ethanol | E | 1.2 | 0.9 | 3.5 | 2.5 | - | 0.4 | 0.4 | 0.4 | 0.4 | **0.3** |
| | F | 7.3 | 6.2 | 18.1 | 13.4 | 3.6 | 2.8 | 3.2 | 2.1 | 2.1 | **1.8** |
| Malonaldehyde | E | 1.7 | 1.5 | 4.8 | 4.6 | - | 0.8 | 0.8 | 0.6 | 0.8 | **0.5** |
| | F | 11.1 | 10.3 | 26.4 | 24.5 | 6.6 | 5.1 | 5.8 | **3.6** | 4.1 | **3.6** |
| Naphthalene | E | 0.9 | 1.2 | 3.8 | 11.3 | - | 0.9 | **0.2** | **0.2** | 0.5 | **0.2** |
| | F | 5.1 | 6.5 | 16.5 | 29.2 | 1.9 | 1.3 | 1.8 | **0.9** | 1.6 | **0.9** |
| Paracetamol | E | 4.0 | 2.9 | 8.5 | 11.5 | - | 1.4 | 1.3 | 1.5 | 1.3 | **0.9** |
| | F | 12.7 | 12.3 | 28.9 | 30.4 | - | 5.9 | 5.8 | 4.9 | 4.8 | **4.0** |
| Salicylic acid | E | 1.8 | 1.8 | 5.6 | 9.2 | - | 0.7 | 0.8 | 0.9 | 0.9 | **0.5** |
| | F | 9.3 | 9.5 | 24.7 | 29.7 | 5.3 | 4.0 | 3.1 | 4.3 | **2.9** | **2.9** |
| Toluene | E | 1.1 | 1.7 | 4.0 | 7.7 | - | 0.3 | 0.3 | 0.4 | 0.5 | **0.2** |
| | F | 6.5 | 8.8 | 17.8 | 24.3 | 2.2 | 1.6 | 1.9 | 1.8 | 1.5 | **1.1** |
| Uracil | E | 1.1 | 0.6 | 3.0 | 5.1 | - | 0.4 | 0.4 | 0.6 | 0.5 | **0.3** |
| | F | 6.6 | 4.2 | 17.6 | 21.4 | 3.8 | 3.1 | 3.2 | **1.8** | 2.1 | 2.0 |

Specifically, to approximate $n$ different positions in $D$-spanning function, PACE without SI needs $v_{\max}n$ channels, while PACE with SI only needs $n$ channels. This is because the SI in the polynomial many-body interaction module enables every single channel in PACE to approximate one position in $D$-spanning functions. The corresponding proof is shown in Appendix B.2.2. The total number of self-interactions equals the max correlation order $v_{\max}$, where $v_{\max}$ corresponds to (body order $-1$). In our implementations, we use three self-interactions because the highest body order we consider in each layer is 4. Although $v_{\max}n$ channels without self-interactions can achieve the same expressiveness theoretically, we find adding these additional self-interactions can significantly enhance the empirical performance.

## 5 Experiments

We conduct experiments on three molecular dynamics simulation datasets, the revised MD17 (rMD17), 3BPA and AcAc datasets. The proposed PACE is trained using these datasets to predict both the invariant energy of the entire molecule and the equivariant forces acting on individual atoms. Among our baselines (Kovács et al., 2021; Christensen et al., 2020; Bartók et al., 2010; Smith et al., 2017; Gasteiger et al., 2021; Batzner et al., 2022; Batatia et al., 2022a; Musaelian et al., 2023; Batatia et al., 2022b), NequIP, BOTNet, Allegro and MACE are all equivariant graph neural networks (GNNs) with rotation order $\ell > 1$. In particular, BOTNet, Allegro, and MACE incorporate many-body interactions, while ACE is a parameterized physical model that does not belong to the class of neural networks. Our experiments are implemented with PyTorch 1.11.0 (Paszke et al., 2019), PyTorch Geometric 2.1.0 (Fey & Lenssen, 2019), and e3nn 0.5.1 (Geiger & Smidt, 2022). In experiments, we train models on a single 11GB Nvidia GeForce RTX 2080Ti GPU and Intel Xeon Gold 6248 CPU.

### 5.1 The rMD17 Dataset

**Dataset.** The rMD17 (Christensen & Von Lilienfeld, 2020) is a benchmark dataset that comprises ten small organic molecular systems. Each molecule in the dataset is accompanied by 1000 3D structures, which were generated through meticulously accurate ab initio molecular dynamic simulations employing density

Table 2: Performance on the 3BPA dataset. Root-mean-square errors (RMSE) are reported for both energy (E) and force (F) predictions, with meV and meV/Å as units, respectively. Standard deviations are calculated over three runs with different seeds. Bold numbers highlight the best performance.

| | | ACE | NequIP | BOTNet | Allegro | MACE | Ours |
|---|---|---|---|---|---|---|---|
| 300K | E | 1.1 | 3.28 (0.12) | 3.1 (0.13) | 3.84 (0.10) | 3.0 (0.2) | **2.4** (0.1) |
| | F | 27.1 | 10.77 (0.28) | 11.0 (0.14) | 12.98 (0.20) | **8.8** (0.3) | 9.1 (0.1) |
| 600K | E | 24.0 | 11.16 (0.17) | 11.5 (0.6) | 12.07 (0.55) | 9.7 (0.5) | **7.9** (0.2) |
| | F | 64.3 | 26.37 (0.11) | 26.7 (0.29) | 29.11 (0.27) | 21.8 (0.6) | **21.4** (0.3) |
| 1200K | E | 85.3 | 38.52 (2.00) | 39.1 (1.1) | 42.57 (1.79) | 29.8 (1.0) | **29.6** (0.4) |
| | F | 187.0 | 76.18 (1.36) | 81.1 (1.5) | 82.96 (2.17) | 62.0 (0.7) | **60.7** (2.0) |
| Dihedral Slices | E | - | 23.3 | - | 16.3 (1.5) | 7.9 (0.6) | **7.6** (0.4) |
| | F | - | 23.1 | - | 20.0 (1.2) | 16.5 (1.7) | **16.0** (0.5) |

Table 3: Performance on the AcAc dataset. Root-mean-square errors (RMSE) are reported for both energy (E) and force (F) predictions, with meV and meV/Å as units, respectively. Standard deviations are calculated over three runs with different seeds. Bold numbers highlight the best performance.

| | | BOTNet | NequIP | MACE | Ours |
|---|---|---|---|---|---|
| 300K | E | 0.89 (0.0) | 0.81 (0.04) | 0.9 (0.03) | **0.73** (0.02) |
| | F | 6.3 (0.0) | 5.90 (0.38) | 5.1 (0.1) | **4.2** (0.1) |
| 600K | E | 6.2 (1.1) | 6.04 (1.26) | 4.6 (0.3) | **3.8** (0.6) |
| | F | 29.8 (1.0) | 27.8 (3.29) | 22.4 (0.9) | **16.9** (0.3) |

functional theory (DFT). These structures capture the diverse conformational space of the molecules and are valuable for studying their quantum properties.

**Setup.** In our experiments, we use officially provided random splits. Next, we use the same splitting seed as MACE to further divide the training set into a training set comprising 950 structures and a validation set comprising 50 structures. Then, we perform our evaluations on the test set with 1000 structures. Training details are provided in Appendix C.2.

**Results.** Table 1 summarizes the performance of our proposed method in comparison to baselines on all ten molecules in the rMD17 dataset. Mean absolute errors (MAE) are employed as the evaluation metric for both energy and force predictions. It is worth noting that our PACE demonstrates state-of-art performance in energy prediction across all molecules. Specifically, we achieved significant improvements of 33.3%, 33.3% and 30.8% on Benzene, Toluene, and Paracetamol, respectively. In terms of force prediction, PACE achieves state-of-the-art performance on nine out of the ten molecules, exhibiting substantial improvements of 26.7%, 16.7% and 15.4% on Toluene, Paracetamol, and Azobenzene, respectively. Besides, we attain the second-best on Uracil. The comparison of algorithm efficiency and the affect from rotation order can be found in Appendix D.2 and D.3.

## 5.2 The 3BPA & AcAc Datasets

**Dataset.** The 3BPA dataset (Kovács et al., 2021) is also generated through molecular dynamic simulations employing Density Functional Theory (DFT). Unlike rMD17, this dataset is specifically focused on a single flexible molecule, namely the 3BPA molecule. 3BPA is characterized by three freely rotating angles, which primarily induce structural changes at varying temperatures. The AcAc dataset (Batatia et al., 2022a) is also generated through molecular dynamic simulation using Density Functional Theory (DFT). Similar to 3BPA, this dataset specifically focuses on a single flexible molecule, Acetylacetone. Given that many real-world applications involve temperature fluctuations, evaluating a model's robustness and extrapolation capabilities under varying temperature conditions is crucial for predicting molecular behavior accurately. 3BPA and AcAc datasets are frequently employed to assess a model's out-of-distribution (OOD) generalization capability to MD geometric distributions across different temperatures for the same molecule.

Table 4: Results of ablation experiments. Compared to PACE, PACE (-EB) removes edge booster in the first layer, and PACE (-SI) uses no additional self-interaction in the polynomial many-body interaction module. Mean absolute errors (MAE) are reported for both energy (E) and force (F) predictions, with meV and meV/Å as units, respectively. Bold numbers highlight the best performance.

|  |  | AcAc 300K | AcAc 600K |
|---|---|---|---|
| PACE | E | **0.73 (0.02)** | **3.8 (0.6)** |
|  | F | **4.2 (0.1)** | **16.9 (0.3)** |
| PACE (-EB) | E | 0.84 (0.04) | **3.8 (0.1)** |
|  | F | 4.9 (0.1) | 18.0 (0.3) |
| PACE (-SI) | E | 0.95 (0.01) | 4.0 (0.2) |
|  | F | 5.7 (0.1) | 21.3 (1.0) |

**Setup.** For both 3BPA and AcAc, We follow MACE to train our model using a training set consisting of 450 structures and a validation set comprising 50 structures. Both the training and validation sets were sampled at 300K. Both 3BPA and AcAc have two test sets sampled at 300K and 600K. 3BPA has another test set sampled at 1200K. Moreover, 3BPA has an additional test set, including optimized conformations with dihedral slices, where two dihedral angles are fixed and the third angle varies from 0 to 360 degrees. This test set probes regions of the potential energy surface (PES) that are distant from the training set. We provide training details in Appendix C.2.

**Results.** Table 2 and Table 3 summarize the performance of our proposed method in the 3BPA and AcAc datasets, respectively. Here, root-mean-square error (RMSE) is used as the evaluation metric. On 3BPA dataset, the proposed PACE shows comparable performance to MACE, while outperforming other baseline methods significantly. On AcAc dataset, our PACE demonstrates significant performance improvement from all baselines. Moreover, we use the model trained on 3BPA to conduct molecular dynamic simulations and the comparison of PACE-generated trajectories and the ground-truth trajectories is shown in Appendix D.1.

### 5.3 Ablation Studies

The first ablation study aims to demonstrate the effectiveness of the proposed edge booster, designed to facilitate a higher polynomial degree. In this experiment, we remove the second tensor product, namely the edge booster, from the first layer of PACE. As indicated in Table 4, this comparison reveals that the edge booster, which enhances expressiveness, indeed imrpoves the model's performance in force field prediction.

The second ablation study is for demonstrating the effectiveness of using different atomic bases $A_{iv}$ for different body orders in many-body interaction. In this experiment, we remove additional self-interaction operations and directly use $A_i$ in symmetric contraction in the polynomial many-body interaction module. The comparison shown in Table 4 justifies that different atomic bases $A_{iv}$ produced by additional self-interactions can improve the expressiveness and model performance without increasing the number of channels.

## 6 Conclusion

In this work, we introduced PACE, a new equivariant network for atomic potential and force field predictions. Our PACE is designed from two perspectives, increasing the ability to approximate more positions in D-spanning functions, as well as expanding the space to cover a greater number of higher degree polynomial equivariant functions. The integration of the edge booster and Atomic Cluster Expansion (ACE) technique with additional self-interactions allows PACE to approximate $SE(3) \times S_n$ equivariant polynomial functions with higher degrees, thereby improving the accuracy in prediction. The comprehensive experimental results and analyses provide compelling evidence for the efficacy of the proposed PACE model. In the future, we aim to advance the development of models capable of approximating polynomial functions of even higher degrees while simultaneously maintaining high computational efficiency.

## Acknowledgements

This work was supported in part by National Science Foundation grant IIS-2243850 and National Institutes of Health grant U01AG070112.

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

## A  Background and Related Work

### A.1  Spherical Harmonics and Irreducible Representation

To encode geometric information of molecules into SE(3)-equivariant features, spherical harmonics are used for its equivariance property. Specifically, we use real spherical harmonic basis functions $Y$ to encode an orientation $\hat{r}_{ij}$ between a node pair. If the molecule is rotated by a rotation matrix $T$ in 3D coordinate system, then we have:

$$\mathbf{q}^\ell Y^\ell(T\hat{r}_{ij}) = (\boldsymbol{D}^\ell(T)\mathbf{q}^\ell)Y^\ell(\hat{r}_{ij}), \tag{24}$$

where $\ell \in [0, L]$ is the degree, $\mathbf{q}^\ell$ with size $2\ell + 1$ denotes coefficients of spherical harmonics, and the Wigner D-matrix $\boldsymbol{D}^\ell(T)$ with size $(2\ell + 1) \times (2\ell + 1)$ specifies the corresponding rotation acting on coefficients $\mathbf{q}^\ell$. In practice, equivariant features are often represented by irreducible representations (irreps), which correspond to the coefficients of real spherical harmonics. Concretely, irreducible representations are formed by the concatenation of multiple type-$\ell$ vectors, where $\ell \in [0, L]$. More detailed explanation about irreducible representations can be found in Liao & Smidt (2023).

### A.2  Tensor Product

Equivariant networks using higher order $\ell > 1$ equivariant features (Thomas et al., 2018; Fuchs et al., 2020; Liao & Smidt, 2023; Batzner et al., 2022; Batatia et al., 2022a;b) predominantly employs tensor products (TP) to preserve higher-order equivariant features. Tensor product operates on irreducible representations $\mathbf{u}$ of rotation order $\ell_1$ and $\mathbf{v}$ of rotation order $\ell_2$, yielding a new irreducible representation of order $\ell_3$ as

$$(\mathbf{u}^{\ell_1} \otimes \mathbf{v}^{\ell_2})_{m_3}^{\ell_3} = \sum_{m_1=-\ell_1}^{\ell_1} \sum_{m_2=-\ell_2}^{\ell_2} C_{(\ell_1,m_1),(\ell_2,m_2)}^{(\ell_3,m_3)} \mathbf{u}_{m_1}^{\ell_1} \mathbf{v}_{m_2}^{\ell_2}, \tag{25}$$

where $C$ denotes the Clebsch-Gordan (CG) coefficients (Griffiths & Schroeter, 2018) and $m \in \mathbb{N}$ denotes the $m$-th element in the irreducible representation. Here, $\ell_3$ satisfies $|\ell_1 - \ell_2| \leq \ell_3 \leq \ell_1 + \ell_2$, and $\ell_1, \ell_2, \ell_3 \in \mathbb{N}$. High-order equivariant 3D GNNs commonly use tensor products on the irreducible representations of neighbor nodes and edges to construct messages as

$$\mathbf{m}_{ij}^{\ell_o} = \sum_{\ell_i,\ell_f} R^{(\ell_i,\ell_f)}(\bar{r}_{ij})Y^{\ell_f}(\hat{r}_{ij}) \otimes \mathbf{x}_j^{\ell_i}, \tag{26}$$

where $|\ell_i - \ell_f| \leq \ell_o \leq \ell_i + \ell_f$, $\mathbf{x}_j$ denotes features of node $j$, and $R$ is a learnable non-linear function that takes the embedding of pairwise distance $\bar{r}_{ij}$ as input.

### A.3  Atomic Cluster Expansion

Molecular potential and force field are crucial physical properties in molecular analysis. To approximate these properties, the atomic cluster expansion (ACE) (Drautz, 2019; Kovács et al., 2021) is used to approximate the atomic potential denoted as

$$E_i(\boldsymbol{\theta}_i) = \sum_j \sum_v c_v^{(1)} \phi_v(\mathbf{r}_{ij}) + \frac{1}{2} \sum_{j_1 j_2} \sum_{v_1 v_2} c_{v_1 v_2}^{(2)} \phi_{v_1}(\mathbf{r}_{ij_1}) \phi_{v_2}(\mathbf{r}_{ij_2})$$
$$+ \frac{1}{3!} \sum_{j_1 j_2 j_3} \sum_{v_1 v_2 v_3} c_{v_1 v_2 v_3}^{(3)} \phi_{v_1}(\mathbf{r}_{ij_1}) \phi_{v_2}(\mathbf{r}_{ij_2}) \phi_{v_3}(\mathbf{r}_{ij_3}) + \cdots, \tag{27}$$

where $\boldsymbol{\theta}_i = (\mathbf{r}_{ij_1}, \cdots, \mathbf{r}_{ij_N})$ denotes the $N$ bonds in the atomic environment, $\phi$ is the single bond basis function and $c$ is the coefficients. The computational complexity of modeling many-body potential increases exponentially with number of neighbors. To reduce the complexity, ACE further makes use of density trick to calculate the atomic energy via atomic base $A_{iv} = \sum_j \phi_v(\mathbf{r}_{ij})$, which has a linear complexity with the number of neighbors, denoted as

$$E_i(\boldsymbol{\theta}_i) = \sum_v c_v^{(1)} A_{iv} + \sum_{v_1 v_2}^{v_1 \geq v_2} c_{v_1 v_2}^{(2)} A_{iv_1} A_{iv_2} + \sum_{v_1 v_2 v_3}^{v_1 \geq v_2 \geq v_3} c_{v_1 v_2 v_3}^{(3)} A_{iv_1} A_{iv_2} A_{iv_3} + \cdots. \tag{28}$$

In this case, the computational complexity of modeling many-body interactions decreases to linear growth with the number of neighbors. With the reduced linearly complexity, many equivariant networks are designed to learn the many-body interactions. In these networks, spherical harmonic functions $Y(\hat{\mathbf{r}}_{ij})$ combined with radial functions $R(\bar{r}_{ij})$ are usually taken as the single bond basis function $\phi_v(\mathbf{r}_{ij})$. Then the aggregated atomic bases $A_{iv}$, typically represented as equivariant irreducible representations in these networks, are combined through tensor product operations to encode the many-body interactions while maintaining equivariance. Note that $v = 0, 1, 2, \ldots$ distinguishes among different basis.

## A.4 Tensor Contraction in Polynomial Many-Body Interaction Module

In the polynomial many-body interaction module of PACE, we apply tensor contraction to different atomic bases for encoding the interaction of multiple nodes around the central node. The tensor contraction which is originally proposed in (Batatia et al., 2022b). However, Figure 2 illustrates the computation of tensor contraction described in Equation (22). This figure demonstrates an example of 4-body interactions with the max correlation order $v_{\max} = 3$, where different atomic bases $A_{i1}$, $A_{i2}$, $A_{i3}$ are fed as inputs. Intuitively, this example is the computation similar to $A_{i1} \otimes A_{i2} \otimes A_{i3} + A_{i1} \otimes A_{i2} + A_{i1}$, where $A_{i1} \otimes A_{i2} \otimes A_{i3}$ approximates 4-body interactions, $A_{i1} \otimes A_{i2}$ approximates 3-body interactions, and $A_{i1}$ approximates 2-body interactions. For easier understanding, this example only has 1 hidden channel, and only shows one output $LM$, that is $L_3 = 0, M_3 = 0$. Generally, the complete final output irreducible representation is the concatenation of all required $L_3 M_3$.

Let's delve deeper into the "contract weights" box and "contract features" box at the top of this figure. The path $\eta[3]$ here denotes a coupled path tuple $(\ell_1, \ell_2, L_2, \ell_3, L_3)$. The purpose of using such coupled paths is for computing consecutive tensor products as a whole instead of applying tensor products one by one. For each path, there is a generalized CG coefficient matrix with shape of $(\ell_{\max}^2, \ell_{\max}^2, \ell_{\max}^2, \ell_{\max}^2, 2 * L_3 + 1)$. Note that $L_3 = 0$ in this example. The top "contrast weights" box executes the first step to take a weighted sum of the generalized CG matrices with learnable weights over the paths. Then, the "contract features" box below applies a dot product on this summed generalized CG matrix and atomic base $A_{i3}$ along the dimension denoted as $\ell_3, m_3$. The output of this step is denoted as $\tilde{a}_{i2,L_2 M_2}^{L_3 M_3}$. Then, $\tilde{a}_{i2,L_2 M_2}^{L_3 M_3}$ is further contracted with $A_{i2}$ and $A_{i1}$ to obtain the final result with similar "contract features" operations. This flow considers the 4-body interactions. To further consider 3-body and 2-body interactions, similar "contract weights" operations are applied over $\eta[2]$ and $\eta[1]$. Then, the results of contracted weights are added to $\tilde{a}_{i2,L_2 M_2}^{L_3 M_3}$ and $\tilde{a}_{i1,L_1 M_1}^{L_3 M_3}$ (represented by arrows pointing from right to left). The summation of generalized CG and the addition of contracted weights to $\tilde{a}_{iv,L_v M_v}^{L_v M_v}$ is for efficient computation of various paths and various body order interactions simultaneously.

As mentioned this example is computing something similar to $A_{i1} \otimes A_{i2} \otimes A_{i3} + A_{i1} \otimes A_{i2} + A_{i1}$ for many-body interactions. However, they are not exactly equivalent, because two tensor products in $A_{i1} \otimes A_{i2} \otimes A_{i3}$ consider the coupled path $\eta$ with generalized CG, which is beyond the original tensor product.

## A.5 Defination

**Definition A.1.** (D-spanning). For $D \in \mathbb{N}_+$, let $\mathcal{F}_{feat}$ be a subset of $\mathcal{C}_G(\mathbb{R}^{3 \times N}, W_{feat}^N)$. We say that $\mathcal{F}_{feat}$ is D-spanning, if there exist $f_1, \cdots, f_K \in \mathcal{F}_{feat}$, such that every polynomial $\mathbb{R}^{3 \times N} \to \mathbb{R}^N$ of degree D which is invariant to translations and equivariant to permutations, can be written as $p(X) = \sum_{k=1}^K \hat{\Lambda}_k (f_k(X))$, where $\Lambda_k : W_{feat} \to \mathbb{R}$ are all linear functionals, and $\hat{\Lambda}_k : W_{feat} \to \mathbb{R}$ are the functions defined by element-wise applications of $\Lambda_k$.

When discussing the expressiveness of equivariant networks, equivariant polynomial functions are commonly used to analyze these networks (Dym & Maron, 2020; Segol & Lipman, 2019). According to Lemma 1 from Dym & Maron (2020), any continuous $G$-equivariant function can be uniformly approximated on compact sets by $G$-equivariant polynomials. Consequently, the ability of a model to approximate equivariant functions increases with its capacity to cover more equivariant polynomials. In this context, a $D$-spanning family refers to a set of functions that span the polynomial function space up to degree $D$. This implies that any $S_n$ permutation equivariant and rotation invariant polynomial function up to degree $D$ can be represented as

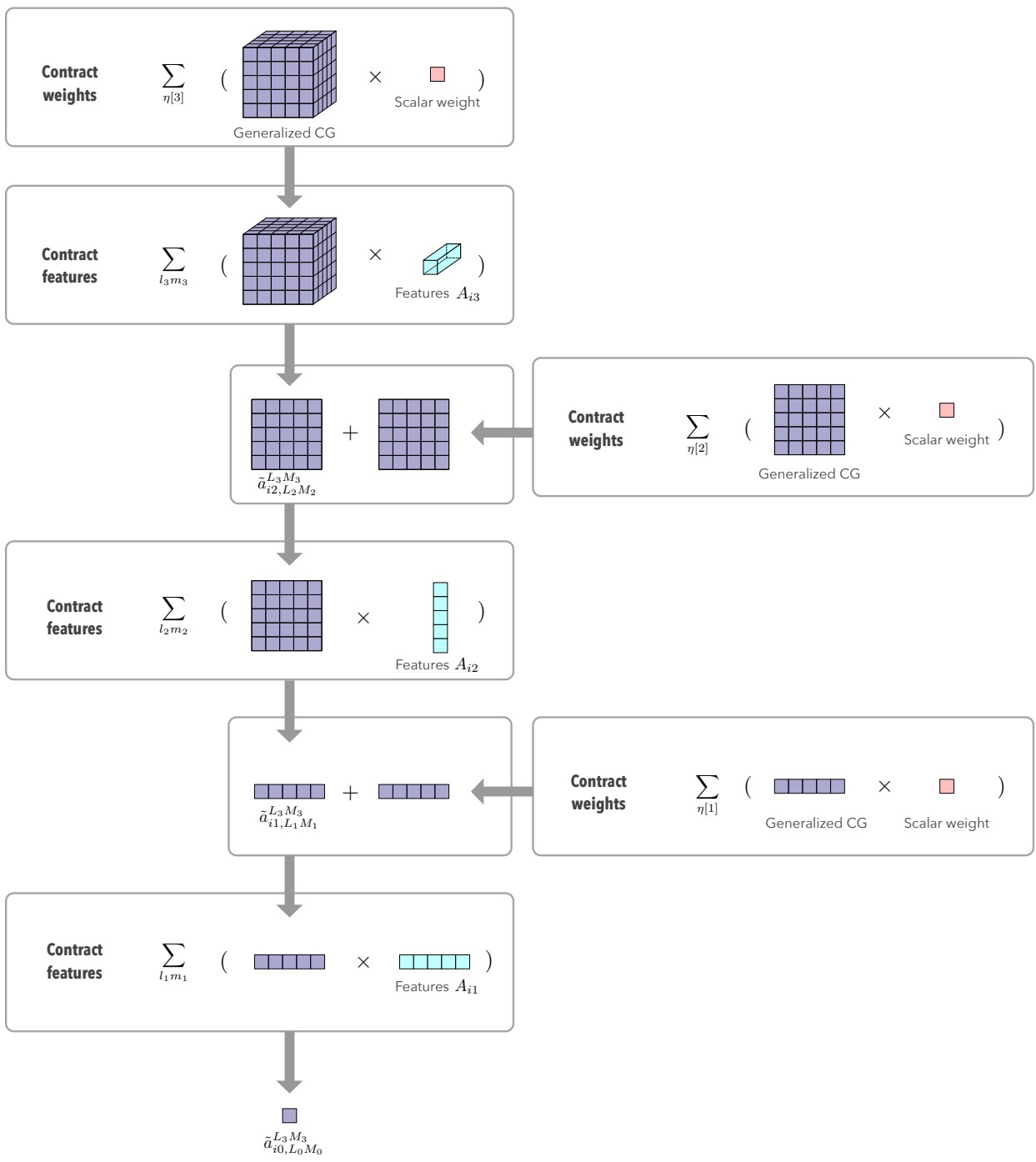

Figure 2: Illustration of tensor contraction in the polynomial many-body interaction module. In this figure, it demonstrates an example of 4-body interactions with $v_{\max} = 3$ and final $L_3 = 0, M_3 = 0$. Note that the contract weights operation learns weighted summation over all paths $\eta[v]$, where $\eta[v] = (\ell_1, \ell_2, L_2, \cdots, \ell_v, L_v)$.

a linear combination of elements within the $D$-spanning family. Therefore, networks that can encompass the $D$-spanning family with a higher $D$ exhibit greater expressiveness.

# B  Theoretical Proof

## B.1  Proof of Theorem 4.1

*Proof.* Since $Q_K^D$ is a D-spanning family, then there exists $f_1, \cdots, f_K \in Q_K^{(\mathbf{t})}$ with $\|\mathbf{t}\|_1 \leq D$, that each polynomial function $p$ with the highest degree no more than $D$ can be represented as the linear combination with linear pooling function $\hat{\Lambda}_k$ on them

$$p = \sum_k w_k * \hat{\Lambda}_k(f_k) = \sum_k w_k \hat{\Lambda}_k(Q_K^{(\mathbf{t_k})}(\boldsymbol{\theta}_i)) \tag{29}$$

$$= \sum_k w_k \sum_P W_P^* Q_K^{(\mathbf{t_k})}(\boldsymbol{\theta}_i)(P), \tag{30}$$

where $P = (p_1, p_2, \cdots, p_K)$ is the position of the entry, and $W_P^*$ is the corresponding weight.

Since $Q^{(\mathbf{t_k})}(\boldsymbol{\theta}_i)(P) = \sum_{\ell m} w_1^{\ell m} \mathrm{irreps}_1^{\ell m}$, the $p$ can be represented as

$$p = \sum_k w_k \sum_P W_P^* \sum_{\ell m} w_{1,\mathbf{t_k},P}^{\ell m} \mathrm{irreps}_{1,\mathbf{t_k},P}^{\ell m} = \sum_k w_k \sum_{\ell m P} w_{1,\mathbf{t_k},P}^{*\ell m} \mathrm{irreps}_{1,\mathbf{t_k},P}^{\ell m} \tag{31}$$

where the $\mathrm{irreps}_{1,\mathbf{t_k},P}^{\ell m}$ can be obtained by various channels and $w_{1,\mathbf{t_k},P}^{*\ell m} = w_{1,\mathbf{t_k},P}^{\ell m} W_P^*$. Therefore, the set of $\mathrm{irreps}_1$ is D-spanning. □

## B.2  Proof of D-spanning irreps in PACE

### B.2.1  Lemmas

**Lemma B.1.** *The linear combination of elements for spherical harmonics with radial basis function $R^\ell(\bar{r}_{ij})\mathbf{Y}_m^\ell(\hat{r}_{ij})$, where $0 \leq \ell \leq \ell_{\max}, \ell \in \mathbb{N}$ and $-\ell \leq m \leq \ell, m \in \mathbb{Z}$, can represent any polynomial function $\mathbf{P}^{\ell_{\max}}(\mathbf{r}_{ij})$.*

*Proof.* When the radial basis function is set to $R^\ell(\bar{r}_{ij}) = \bar{r}_{ij}^\ell, \ell \in \mathbb{N}_+$, real spherical harmonics with radial basis function $R^\ell(\bar{r}_{ij})\mathbf{Y}_m^\ell(\hat{r}_{ij})$ represent homogeneous polynomials $\mathbf{P}(\hat{r}_{ij})$ with degree $\ell$. Moreover, by using multi-layer perceptrons (MLPs) $R^0(\bar{r}_{ij})$, it can approximate any polynomial with respect to the pairwise distance $\bar{r}_{ij}$ (Hornik, 1991; Hornik et al., 1989). We first define the space of homogeneous polynomials $\mathbb{H}_n(\mathbb{R}^3)$, and space of harmonic homogeneous polynomials $\mathbb{Y}_n(\mathbb{R}^3)$. When we constrain the function to be on $\mathbb{S}^2$, $\mathbb{Y}_n \equiv \mathbb{Y}(\mathbb{S}^2)$ and is composed of spherical harmonics. From the Lemma 4.1 in Atkinson & Han (2012), a homogeneous polynomial $p(\mathbf{x}) \in \mathbb{H}_n$ can be decomposed as $p(\mathbf{x}) = h(\mathbf{x}) + |\mathbf{x}|^2 q(\mathbf{x})$, where $h \in \mathbb{Y}_n$ and $q \in \mathbb{H}_{n-2}$ for $n \geq 2$. When considering polynomial functions constrained to $\mathbb{S}^2$, we have $p(\frac{\mathbf{x}}{|\mathbf{x}|}) = h(\frac{\mathbf{x}}{|\mathbf{x}|}) + q(\frac{\mathbf{x}}{|\mathbf{x}|})$, where h is in the space spanned by spherical harmonics of degree $n$. The dimension of homogeneous polynomials is $\frac{(n+1)(n+2)}{2}$. For $n = 1$, the dimension of homogeneous polynomials space is 3, and the dimension of spherical harmonics space is also 3. For $n = 2$, the dimension of homogeneous polynomial space is 6, and the dimension of spherical harmonic space is 5. Then, with the constrain $x^2 + y^2 + z^2 = 1$ denoting the input $(x, y, z)$ on $\mathbb{S}^2$, which is independent with spherical harmonics $Y^{\ell=2}(\hat{r})$. Therefore, homogeneous polynomials on $\mathbb{S}^2$ can be linear combination of spherical harmonics for $n = 2$. Thus, $q(\frac{\mathbf{x}}{|\mathbf{x}|})$ can be expressed as a linear combination of spherical harmonics when $q \in \mathbb{H}_n, n = 1, 2$. Then, we can deduce that a homogeneous polynomial function $q(\frac{\mathbf{x}}{|\mathbf{x}|})$ can be represented as a linear combination of spherical harmonics in $\mathbb{S}^2$. Since a homogeneous polynomial function of degree $n$ can be represented by $q(\mathbf{x}) = |x|^n q(\frac{x}{|x|})$, where $q \in \mathbb{H}_n$, it can be represented by spherical harmonics with radial function to encode the distance. Above all, spherical harmonics with radial basis $R^\ell(\bar{r}_{ij})\mathbf{Y}_m^\ell(\hat{r}_{ij})$ can represent the polynomial function $\mathbf{P}^{\ell_{\max}}(\mathbf{r}_{ij})$.

□

**Lemma B.2.** *If $\mathrm{irreps}_1$ can represent $Q^{\mathbf{t}_1}(\boldsymbol{\theta}_i)$, $\mathrm{irreps}_2$ can represent $Q^{\mathbf{t}_2}(\boldsymbol{\theta}_i)$ and their tensor product output $\mathrm{irreps}_3$ can represent $Q^{(\mathbf{t}_1,\mathbf{t}_2)}(\boldsymbol{\theta}_i)$.*

*Proof.* With the aggregated node features with $Q^{\mathbf{t_1}} = \sum_{j_1} \mathbf{r}_{ij}^{\otimes t_1}$ for tensor representation format, and the value at position $P_1 = (p_{11}, \cdots, p_{iL})$ can be represented by linear combination of the elements in $\text{irreps}_1$, shown as

$$Q^{\mathbf{t_1}}(P_1) = \sum_{\ell_1 m_1} w_1^{\ell_1 m_1} \text{irreps}_1^{\ell_1 m_1}, \tag{32}$$

When the tensor product of $\text{irreps}_1$ and $\text{irreps}_2$ is $\text{irreps}_3$, the representation of $\text{irreps}_3$ is denoted as

$$\text{irreps}_3^{\ell_3(\ell_1,\ell_2)m_3} = \sum_{m_1 m_2} C_{\ell_1 m_1, \ell_2 m_2}^{\ell_3 m_3} \text{irreps}_1^{\ell_1 m_1} \text{irreps}_2^{\ell_2 m_2} \tag{33}$$

When the $Q^{\mathbf{t_2}}(P_2)$ can be linear combination of elements in $\text{irreps}_2$, then

$$\begin{aligned}
Q^{(\mathbf{t_1},\mathbf{t_2})}(P_1, P_2) &= \Big(\sum_{\ell_1 m_1} w_1^{\ell_1 m_1} \text{irreps}_1^{\ell_1 m_1}\Big)\Big(\sum_{\ell_2 m_2} w_2^{\ell_2 m_2} \text{irreps}_2^{\ell_2 m_2}\Big) \\
&= \sum_{\ell_1 m_1 \ell_2 m_2} w_1^{\ell_1 m_1} w_2^{\ell_2 m_2} \text{irreps}_1 \text{irreps}_2
\end{aligned} \tag{34}$$

Then when the linear combination of the $\text{irreps}_3$ is shown as

$$\begin{aligned}
Q'^{(\mathbf{t_1},\mathbf{t_2})}(P_1, P_2) &= \sum_{\ell_3(\ell_1,\ell_2)m_3} w_3^{\ell_3(\ell_1,\ell_2)m_3} \text{irreps}_3^{\ell_3(\ell_1,\ell_2)m_3} \\
&= \sum_{\ell_3(\ell_1,\ell_2)m_3} w_3^{\ell_3(\ell_1,\ell_2)m_3} \sum_{\ell_1 m_1 \ell_2 m_2} C_{\ell_1 m_1, \ell_2 m_2}^{\ell_3 m_3} \text{irreps}_1^{\ell_1 m_1} \text{irreps}_2^{\ell_2 m_2} \\
&= \sum_{\ell_1 m_1 \ell_2 m_2} \text{irreps}_1^{\ell_1 m_1} \text{irreps}_2^{\ell_2 m_2} \sum_{l_3(\ell_1,\ell_2)m_3} C_{\ell_1 m_1, \ell_2 m_2}^{\ell_3 m_3} w_3^{\ell_3(\ell_1,\ell_2)m_3}
\end{aligned} \tag{35}$$

When $w_3^{\ell_3(\ell_1,\ell_2)m_3} = \sum_{\ell_1 m_1, \ell_2 m_2} C_{\ell_1 m_1, \ell_2 m_2}^{\ell_3 m_3} w_1^{\ell_1 m_1} w_2^{\ell_2 m_2}$, we have

$$\begin{aligned}
&\sum_{\ell_1 m_1, \ell_2 m_2} C_{\ell_1 m_1, \ell_2 m_2}^{\ell_3 m_3} w_3^{\ell_3(\ell_1,\ell_2)m_3} \\
&= \sum_{\ell_1 m_1, \ell_2 m_2} C_{\ell_1 m_1, \ell_2 m_2}^{\ell_3 m_3} \sum_{\ell_3(\ell_1,\ell_2)m_3} C_{\ell_1 m_1, \ell_2 m_2}^{\ell_3 m_3} w_1^{\ell_1 m_1} w_2^{\ell_2 m_2} \\
&= \sum_{\ell_1 m_1, \ell_2 m_2} w_1^{\ell_1 m_1} w_2^{\ell_2 m_2} \sum_{\ell_3(\ell_1,\ell_2)m_3} \Big(C_{\ell_1 m_1, \ell_2 m_2}^{\ell_3 m_3} C_{\ell_1 m_1, \ell_2 m_2}^{\ell_3 m_3}\Big) \\
&= \sum_{\ell_1 m_1, \ell_2 m_2} w_1^{\ell_1 m_1} w_2^{\ell_2 m_2}
\end{aligned} \tag{36}$$

Therefore,

$$\begin{aligned}
Q'^{(\mathbf{t_1},\mathbf{t_2})}(P_1, P_2) &= \sum_{\ell_1 m_1, \ell_2 m_2} \text{irreps}_1^{\ell_1 m_1} \text{irreps}_2^{\ell_2 m_2} \sum_{\ell_1 m_1, \ell_2 m_2} w_1^{\ell_1 m_1} w_2^{\ell_2 m_2} \\
&= \sum_{\ell_1 m_1, \ell_2 m_2} \text{irreps}_1^{\ell_1 m_1} \text{irreps}_2^{\ell_2 m_2} w_1^{\ell_1 m_1} w_2^{\ell_2 m_2} \\
&= Q^{(\mathbf{t_1},\mathbf{t_2})}(P_1, P_2)
\end{aligned} \tag{37}$$

Above all, the $\text{irreps}_3$ can represent $Q^{(\mathbf{t_1},\mathbf{t_2})}(P_1, P_2)$. $\qquad\square$

### B.2.2 Equivariant Polynomial Functions for Polynomial Symmetric Contraction

Considering tensor product, the path can be represented as $\eta[2] = (\ell_1, \ell_2, L_2)$, we have $\text{irreps}_1$ and $\text{irreps}_2$ to represent $Q^{\mathbf{t_1}}(P_1)$ and $Q^{\mathbf{t_2}}(P_2)$ with $\|\mathbf{t_1}\|_1 \leq v, \|\mathbf{t_2}\|_1 \leq v$, respectively. Then with Lemma B.2, the output

irreps can represent $Q^{(\mathbf{t_1},\mathbf{t_2})}(P_1, P_2)$. Note that there might be multiple channels for the same rotation order $L_2$, and we use $L_2(\ell_1, \ell_2)$ to distinguish them. Meanwhile, the weighted sum over these equivariant irreducible representations can also achieve representativity.

For the path $\eta[v] = (\ell_1, \ell_2, L_2, \cdots, \ell_v, L_v)$, the $\text{irreps}_{\eta[v-1]}$ can represent $Q^{(\mathbf{t_1},\mathbf{t_2},\cdots,\mathbf{t_{v-1}})}$, and the $\text{irreps}_v$ can represent $Q^{\mathbf{t_v}}$.

$$\text{irreps}_3^{L'_v M_v} = \sum_{M_{v-1} m_v} C_{L'_{v-1} M_{v-1}, \ell_v m_v}^{L_v M_v} \text{irreps}_1^{L'_{v-1} M_{v-1}} \text{irreps}_2^{\ell_v m_v}, \tag{38}$$

where $L'_v = L_v(\eta[v-1], \ell_v)$ and $L'_{v-1} = L_{v-1}(\eta[v-2], \ell_{v-1})$. Then, in this case, we extend the proof of Lemma B.2 and prove the

$$
\begin{aligned}
Q^{'(\mathbf{t_1},\mathbf{t_2},\mathbf{t_v})}(P_1, P_2, \cdots P_v) &= \sum_{L'_v m_3} w_3^{L'_v M_v} \text{irreps}_3^{L'_v M_v} \\
&= \sum_{L'_v M_v} w_3^{L'_v M_v} \sum_{L'_{v-1} M_v, \ell_v m_v} C_{L'_{v-1} M_v, \ell_v m_v}^{\ell_3 m_3} \text{irreps}_1^{L'_{v-1} M_v} \text{irreps}_2^{\ell_v m_v} \\
&= \sum_{L'_{v-1} M_v \ell_v m_v} \text{irreps}_1^{L'_{v-1} M_v} \text{irreps}_2^{\ell_v m_v} \sum_{L'_{v-1} M_v} C_{L'_{v-1} M_v, \ell_v m_v}^{L'_{v-1} M_v} w_3^{L'_v M_v}
\end{aligned}
\tag{39}
$$

Then we take $w_3^{L'_v M_v} = \sum_{\ell_1 m_1, \ell_2 m_2} C_{L'_{v-1} M_{v-1}, \ell_v m_v}^{L_v M_v} w_1^{L'_{v-1} M_{v-1}} w_2^{\ell_v m_v}$, and with similar procedure to Equation (37), we can derive that

$$
\begin{aligned}
&Q^{'(\mathbf{t_1},\mathbf{t_2},\cdots,\mathbf{t_v})}(P_1, P_2, \cdots P_v) \\
&= \sum_{L'_{v-1} M_v \ell_v m_v} w_1^{L'_{v-1} M_{v-1}} \text{irreps}_1^{L'_{v-1} M_v} w_2^{\ell_v m_v} \text{irreps}_2^{\ell_v m_v} \\
&= Q^{(\mathbf{t_1},\mathbf{t_2},\cdots,\mathbf{t_{v-1}})}(P_1, P_2, \cdots P_{v-1}) Q^{\mathbf{t_v}}(P_v)
\end{aligned}
\tag{40}
$$

Above all, the output of equivariant base can represent $Q^{(\mathbf{t_1},\mathbf{t_2},\cdots,\mathbf{t_v})}(\boldsymbol{\theta}_i)$ with $\|\mathbf{t}_j\|_1 \leq v, \forall j \in [v]$. Since it can select any $(\mathbf{t_1},\mathbf{t_2},\cdots,\mathbf{t_v})$ within $\|(\mathbf{t_1},\mathbf{t_2},\cdots,\mathbf{t_v})\|_1 \leq v$, the constructed D-spanning family $Q_K^D$ with $K = v$ and $D = v$. Then, the set of the output equivariant features is also a D-spanning family with $D = v$.

Specifically, in the update module in the first layer, it uses symmetric contraction with correlation order $v_{\max} = 3$. Since the aggregated messages follow the form $Q_{K=1}^{(\mathbf{t})} = \sum_j \mathbf{r}_{ij}^{\otimes t_1}$ with $t_1 \leq \ell_{\max} * N_{\text{boost}}$ with $N_{\text{boost}} = 2$ and $v_{\max} = 3$. Then, the updated features after the symmetric contraction is $Q_{K=3}^{(\mathbf{t})} = \sum_{j_1, j_2, j_3 \in \mathcal{N}_i} \mathbf{r}_{ij_1}^{\otimes t_1} \otimes \mathbf{r}_{ij_2}^{\otimes t_2} \otimes \mathbf{r}_{ij_3}^{\otimes t_3}, t_1, t_2, t_3 \leq 6$. Therefore, it covers all the possible $\mathbf{t}$ with $\|\mathbf{t}\| \leq 3$, and then represent the $D$-spanning set with $D = 3$.

## C Implementation Details

### C.1 Data Preprocessing

The raw data and splits for the rMD17 dataset are downloaded from `https://figshare.com/articles/dataset/Revised_MD17_dataset_rMD17_/12672038`. The raw data is provided in Numpy .npz format, with energies in units of kcal and forces in units of kcal/Å. We convert the units for energies and forces to eV and eV/Å, and save the data in .xyz format. The 3BPA and AcAc datasets, already in .xyz format with energies in eV and forces in eV/Å, are directly downloaded from `https://github.com/davkovacs/BOTNet-datasets/tree/main`.

Next, we extract nuclear charges, conformation coordinates, energies, and forces from the .xyz files and encapsulate all this information into a PyTorch Geometric dataset. Subsequently, we perform another pre-processing step to obtain one-hot encoding for each atom and construct a molecular graph using a radius

cutoff. It is important to note that we train a separate model for each molecule. Consequently, the dimension of the one-hot encoding for nodes depends on the total number of atomic types present in the molecule.

Following this, we compute several statistics of the dataset. First, we calculate the average energy using the training set. Then, we determine the mean per-atom energy errors. Second, we calculate the standard deviation of per-atom forces. Third, we compute the average number of neighboring nodes. The processed training dataset, along with the pre-computed statistics, is then used to train our model. This pre-processing workflow primarily follows the baseline method, MACE, without incorporating additional features.

## C.2 Experiment Details

For all molecules in, we use 2 GNN layers with 256 hidden channels and $\ell_{\max} = 3$. For multi-layer perceptions (MLPs), SiLU is used as nonlinearity. During training, we train each model for 5000 epochs using a batch size of 5 and a learning rate of 0.01. Besides, we use Adam-AMSGrad optimizer with default parameters of $\beta_1 = 0.9$, $\beta_2 = 0.999$, $\epsilon = 10^{-8}$, and without weight decay. Moreover, an exponential moving average with a weight of 0.99 is used in training.

For rMD17, edges are built for pairwise distances within a radius cutoff. Edge features are initiated by either Exponential Bernstein radial basis functions (Unke et al., 2021b) or bessel functions with a smoothed polynomial cutoff (Gasteiger et al., 2020). Table 5 shows our options for molecules in rMD17 dataset. The training loss considers both energy and force with a weight ratio of 9: 1000. In addition, we use a rotation order of 3 for the hidden irreducible representations outputted by PACE's polynomial many-body interaction module. For both 3BPA and AcAc, edges are built using a radius cutoff of 5 Å, and edge features are initiated with 4 Bessel basis functions. The ratio of energy and force in loss is 15:1000. Moreover, the rotation order of the hidden irreducible representations is 2.

Table 5: Model architectural hyperparameters for rMD17.

|  | Aspirin | Azobenzene | Benzene | Ethanol | Malonaldehyde | Naphthalene | Paracetamol | Salicyclic acid | Toluene | Uracil |
|---|---|---|---|---|---|---|---|---|---|---|
| Edge embedding | Bessel | Bessel | Bessel | EBRadial | EBRadial | Bessel | Bessel | EBRadial | Bessel | Bessel |
| # of basis | 6 | 4 | 8 | 20 | 12 | 8 | 4 | 10 | 8 | 4 |
| Radius (Å) | 5 | 5 | 5 | 5 | 6 | 5 | 5 | 5 | 5 | 6 |

## C.3 Pseudocode of PACE

In this subsection, we provide pseudocode for implementing our PACE. The initial node features are embedded based on atomic types, edge directions are encoded by spherical harmonics, and edge lengths are encoded by radial basis functions. These node and edge features are then fed into the first message passing layer. In this layer, an edge booster is first applied to produce boosted messages along the edges. These boosted messages are then aggregated around the central node. Next, the aggregated message and initial node features are processed by a polynomial many-body interaction module, followed by a skip connection with self-interaction to update the node features. In the second message passing layer, a tensor product is applied to the updated node features and the boosted messages obtained from the first layer to update edge messages. As in the first layer, edge messages are aggregated and further processed by the polynomial many-body interaction module to update the node features. Finally, the total energy and forces acting on atoms are calculated based on the updated node features and atom coordinates.

The first key contribution of our method is the use of an edge booster to increase the number of the higher-degree polynomial functions that our model can approximate. The edge booster is implemented in the first PACE layer using two consecutive tensor products. Initially, a tensor product is applied to the edge spherical harmonics and the initial node features to generate the boosted message with $N_{\text{boost}} = 1$. Subsequently, another tensor product is applied to the edge spherical harmonics and the previously obtained boosted message, resulting in a boosted message with $N_{\text{boost}} = 2$. The final boosted message is obtained by concatenating these boosted messages with $N_{\text{boost}} = 1$ and $N_{\text{boost}} = 2$. This boosted message is used in both PACE layers without repetitive computation, taking computational costs into consideration. The second key

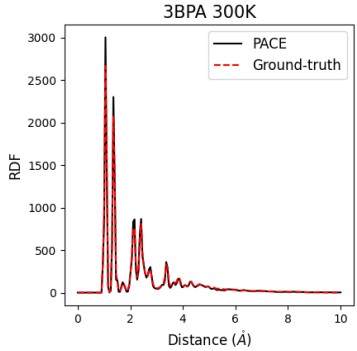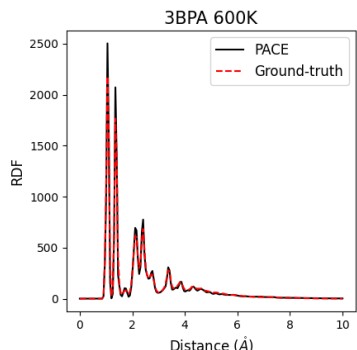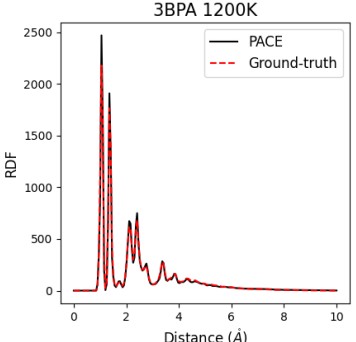

Figure 3: Illustration of radial distribution functions (RDF) of MD trajectories. Values are averaged over five MD simulations with five initial molecular structures. The shell thickness $dr = 0.05$ is used.

contribution of our method is the incorporation of additional self-interactions in the polynomial many-body interaction module. These self-interactions produce different atomic bases, which serve as inputs for symmetric contraction. The function of these self-interactions is to enhance the model's ability to approximate more positions in $D$-spanning functions.

In our implementation, the MLP applied to invariant features consists of a 2-layer non-linear transformation with SiLu activation. Detailed information about the tensor product and self-interaction applied to equivariant features can be found in the TFN (Thomas et al., 2018) and e3nn (Geiger & Smidt, 2022) libraries. For a more comprehensive explanation of the algorithm used for symmetric contraction, please refer to MACE (Batatia et al., 2022b).

# D  Additional Analysis

## D.1  Molecular Dynamic Simulation

To further assess the ability of PACE to simulate realistic structures and dynamics, we conduct Molecular Dynamics (MD) simulations on the 3BPA dataset using PACE. These simulations are implemented with the Langevin dynamics provided by the ASE library, using 1000 timesteps with 1 fs for each timestep. We randomly select an initial structure from the testing set to generate MD trajectories with PACE. Correspondingly, we extract the ground truth MD trajectory, starting from the same initial structure, from the testing set. The total radial distribution functions (RDF) of the ground truth MD trajectories and PACE-generated MD trajectories are visualized in Figure 3. The results of the MD simulations further affirm that our PACE model not only achieves state-of-the-art (SOTA) performance in force field prediction but also holds practical value in realistic simulations.

## D.2  Algorithm Efficiency

### D.2.1  Time Complexity of PACE

Table 6 shows the time complexity for several key components of existing equivariant model architectures. In this table, $C$ is the number of channels, $L$ is the maximum rotation order of the equivariant features, and $v$ denotes the correlation order. Our PACE model requires three tensor products (TPs) to obtain the edge features and two polynomial many-body interaction modules on the aggregated node features. Each tensor product has a time complexity of $O(CL^6)$. Compared to the many-body interaction module used in MACE with a time complexity of $O(C^2L + CL^{4v+2})$, our polynomial many-body interaction module has a time complexity of $O(NCL^6 + ECL^{4v+2})$, where $N$ denotes the number of nodes, $E$ denotes the number of edges.

---

**Algorithm 1** Code sketch of PACE network. The PACE architecture comprises embedding layers, two distinct message passing (MPNN) layers, and an output layer. The proposed edge booster is integrated into the first MPNN layer to generate the boosted edge message $\mathbf{m}_{ij}^1$. In the polynomial many-body interaction module, we utilize additional self-interactions to produce different atomic bases $A_{iv}$.

---

**Require:** $Z_i, |\mathcal{N}(i)|, \mathbf{p}_i, \hat{r}_{ij}, \bar{r}_{ij}, \bar{E}$ ▷ One-hot encoding of node by atomic types $Z_i$, the number of neighbor nodes $|\mathcal{N}(i)|$, node positions $\mathbf{p}_i$, edge orientation vector $\hat{r}_{ij}$, edge distance $\bar{r}_{ij}$, the averaged total energy of the training set $\bar{E}$.

 

    **function** POLYNOMIALMANYBODYINTERACTION($A_i, \mathbf{x}_i^0$)
        $A_{i1} \leftarrow$ SELFINTERACTION($A_i$))                            ▷ Different atomic bases
        $A_{i2} \leftarrow$ SELFINTERACTION($A_i$))
        $A_{i3} \leftarrow$ SELFINTERACTION($A_i$))
        $\tilde{a} \leftarrow$ SYMMETRICCONTRACTION($A_{i1}, A_{i2}, A_{i3}, \mathbf{x}_i^0$)
        **return** $\tilde{a}$
    **end function**
    **function** MPNNLAYER_1($Sph_{ij}, rbf_{ij}, \mathbf{x}_i^0, \mathbf{x}_j^0$)
        $w_{1,ij} \leftarrow$ MLP($rbf_{ij}$)
        $w_{2,ij} \leftarrow$ MLP($rbf_{ij}$) + MLP($\mathbf{x}_i^0 \parallel \mathbf{x}_j^0$)
        $\mathbf{m}_{ij,1}^1 \leftarrow Sph_{ij} \otimes_{w_{1,ij}}$ MLP($\mathbf{x}_i^0 \parallel \mathbf{x}_j^0$)           ▷ Boosted message with $N_{\text{boost}} = 1$.
        $\mathbf{m}_{ij,2}^1 \leftarrow Sph_{ij} \otimes_{w_{2,ij}} \mathbf{m}_{ij,1}^1$                 ▷ Boosted message with $N_{\text{boost}} = 2$
        $\mathbf{m}_{ij}^1 \leftarrow (\mathbf{m}_{ij,1}^1 \parallel \mathbf{m}_{ij,2}^1)$                   ▷ Concatenated boosted message
        $A_i^1 \leftarrow \frac{1}{|\mathcal{N}(i)|} \sum_{j \in \mathcal{N}(i)} \mathbf{m}_{ij}^1$                     ▷ Aggregated message
        $\tilde{a}_i^1 \leftarrow$ POLYNOMIALMANYBODYINTERACTION($A_i^1, \mathbf{x}_i^0$)
        $\mathbf{x}_i^1 \leftarrow \tilde{a}_i^1 +$ SELFINTERACTION($\mathbf{x}_i^0$)                  ▷ Skip connection
        **return** $\mathbf{x}_i^1, \mathbf{m}_{ij}^1$
    **end function**
    **function** MPNNLAYER_2($\mathbf{m}_{ij}^1, rbf_{ij}, \mathbf{x}_i^0, \mathbf{x}_j^0, \mathbf{x}_i^1, \mathbf{x}_j^1$)
        $w_{3,ij} \leftarrow$ MLP($rbf_{ij}$) + MLP($\mathbf{x}_i^0 \parallel \mathbf{x}_j^0$)
        $\mathbf{m}_{ij}^2 \leftarrow \mathbf{m}_{ij}^1 \otimes_{w_{3,ij}} \mathbf{x}_j^1$,                         ▷ Tensor product
        $A_i^2 \leftarrow \frac{1}{|\mathcal{N}(i)|} \sum_{j \in \mathcal{N}(i)} \mathbf{m}_{ij}^2$                     ▷ Aggregated message
        $\tilde{a}_i^2 \leftarrow$ POLYNOMIALMANYBODYINTERACTION($A_i^2, \mathbf{x}_i^0$)
        $\mathbf{x}_i^2 \leftarrow \tilde{a}_i^2 +$ SELFINTERACTION($\mathbf{x}_i^1$)                  ▷ Skip connection
        **return** $\mathbf{x}_i^2$
    **end function**
    $\mathbf{x}_i^0 \leftarrow$ NODEEMBEDDING($Z_i$)                   ▷ Embedding of node via atomic type
    $Sph_{ij} \leftarrow$ SPHERICALHARMONICEMBEDDING($\hat{r}_{ij}$)         ▷ Embedding of edge orientation
    $rbf_{ij} \leftarrow$ RADIALBASISEMBEDDING($\bar{r}_{ij}$)            ▷ Embedding of edge length
    $\mathbf{x}_i^1, \mathbf{m}_{ij}^1 \leftarrow$ MPNNLAYER_1($Sph_{ij}, rbf_{ij}, \mathbf{x}_i^0, \mathbf{x}_j^0$)
    $\mathbf{x}_i^2 \leftarrow$ MPNNLAYER_2($\mathbf{m}_{ij}^1, rbf_{ij}, \mathbf{x}_i^0, \mathbf{x}_j^0, \mathbf{x}_i^1, \mathbf{x}_j^1$)
    $E \leftarrow \bar{E} + \sum_{i=1}^N (\text{MLP}(\mathbf{x}_{i,\ell m=00}^1) + \text{MLP}(\mathbf{x}_{i,\ell m=00}^2))$     ▷ Molecule-level energy
    $\mathbf{f}_i = -\frac{\partial E}{\partial \mathbf{p}_i}$                                               ▷ Atom-level force

---

### D.2.2   Training Time and Memory of PACE

Table 7 presents a comparative analysis of training time and memory consumption between our proposed PACE method and the baseline methods. Results are reported in seconds per epoch and MB as units. A consistent batch size of 5 is used for each method, with average times calculated over 10 epochs, where validation occurs once every 2 epochs. In these experiments, Allegro is configured with 5 layers and a rotation order of 3, and NequIP with 3 layers and a rotation order of 3. MACE is configured with 2 layers and a rotation order of 2. PACE is configured as reported in Appendix C.2. The results show that compared to MACE, PACE offers enhanced expressiveness and superior performance, albeit with a justifiable increase in

Table 6: Time complexity of components used in equivariant model architectures. Here, $C$ is the number of channels, $L$ is the maximum rotation order of the equivariant features, and $v$ denotes the correlation order.

| Model component | Time Complexity |
|---|---|
| Self-interaction | $O(C^2 L)$ |
| Tensor product | $O(CL^6)$ |
| Polynomial many-body Interaction | $O(vC^2 L + CL^{4v+2})$ |
| MACE many-body interaction | $O(C^2 L + CL^{4v+2})$ |

Table 7: Training time and memory consumption, with sec/epoch and MB as units, respectively.

| | | NequIP | Allegro | MACE | Ours |
|---|---|---|---|---|---|
| Paracetamol | Training time | 130 | 50.3 | 29.5 | 45 |
| | Memory | 3511 | 10211 | 3239 | 4406 |
| Toluene | Training time | 110 | 35.8 | 23.5 | 35 |
| | Memory | 3507 | 3595 | 2540 | 3394 |

computational costs. In contrast to NequIP and Allegro, PACE stands out for its combined advantages in both performance and efficiency.

### D.2.3 Cost Analysis of Edge Booster

As summarized in Section 4.5, the proposed edge booster (EB) module enhances our PACE model's ability to approximate higher-degree polynomial functions. Instead of integrating the EB module into each PACE layer, we employ it solely in the first layer of our PACE network. The edge-boosted message generated by the EB is then utilized to replace the original spherical harmonics in every PACE layer. Moreover, the EB module only takes spherical harmonics and atomic types of nodes as inputs without considering the updated node features. This design choice aims to minimize the computational cost associated with approximating more higher-degree polynomial functions. Accordingly, in Table 8 we provide a quantitative analysis of the computational cost incurred by our edge booster.

This analysis of runtime and memory consumption is conducted using the AcAc dataset. Results are reported in seconds per epoch and megabytes (MB). A batch size of 5 is used, and the average time is calculated over 10 epochs, with validation occurring once every 2 epochs. Our PACE model comprises 2 layers. In the first experiment, we remove the EB module and use the original spherical harmonics for tensor products in both PACE layers. The second experiment employs the EB module only in the first layer, which corresponds to the architecture of our PACE. In the third experiment, we add an additional EB module in the second layer. Here, the second-layer EB module uses the updated node features output by the first layer, instead of the node embeddings based on atomic types. The results demonstrate that the EB module with two consecutive tensor product operations is computationally expensive. Compared to incorporating the EB module into both layers, the design of using only one EB module in the first layer, and without considering updated node features, can efficiently reduce the time and memory cost by 30.5% and 9.7%, respectively.

### D.3 Rotation Order

As described in Appendix C.2, for the rMD17 dataset, PACE utilizes a rotation order of 3 for the hidden irreducible representations outputted by its polynomial many-body interaction module, contrasting with MACE's use of a rotation order of 2 for encoding many-body interactions. Hence, we conduct further experiments to determine if the enhanced performance of PACE is solely due to this increased rotation order. The results in Table 9 indicate that while the higher rotation order of hidden representations does contribute to improved model performance, it is not the sole factor in PACE's superior performance.

Table 8: Analysis of computational cost introduced by the edge booster (EB) module on the AcAc dataset. Training time and memory consumption are reported with sec/epoch and MB as units, respectively.

|  | No EB | EB in first layer | EB in both layers |
|---|---|---|---|
| Training time | 12.1 | 17.8 | 25.6 |
| Memory | 2080 | 3076 | 3408 |

Table 9: Results of experiments on rotation order of hidden irreducible representation of polynomial many-body interaction module. PACE uses $\ell_{\max} = 3$ for rMD17 dataset. Mean absolute errors (MAE) are reported for both energy (E) and force (F) predictions, with meV and meV/ Å as units, respectively. Bold numbers highlight the best performance.

|  |  | Ethanol | Toluene |
|---|---|---|---|
| PACE | E | **0.3** | **0.2** |
|  | F | **1.8** | **1.1** |
| PACE (hidden $\ell_{\max} = 2$) | E | **0.3** | **0.2** |
|  | F | 2.1 | **1.1** |

## D.4 Hyperparameter Exploration

In this subsection, we conduct experiments on the AcAc dataset to investigate our model's sensitivity to different hyperparameter settings. Figure 4 presents the performance curves corresponding to various hyperparameter choices. From top to bottom, the results pertain to four hyperparameters: radius cutoff, number of channels, number of radial basis functions (RBFs), and maximum rotation order. Each data point represents the average performance over three runs with different random seeds.

It is widely believed that the radius cutoff can impact model performance, as it determines the connectivity within the 3D molecular graph. Our results indicate that a cutoff of approximately 4-5 Å yields the best performance for our model. Next, the results in the second row indicate that the model performance remains relatively stable with the number of channels ranging from 100 to 300. Having too few channels reduces the model's capacity, while having too many channels does not provide additional benefits. The results in the third row demonstrate that the model's performance remains relatively stable, with optimal performance occurring when using around 4 radial basis functions. Using too few RBFs will reduce the impact of edge distance. In contrast, the last row shows that the maximum rotation order for spherical harmonics and irreducible representations has a significant impact on model performance, which is consistent with trends observed in previous studies (Batzner et al., 2022; Batatia et al., 2022b). However, it is important to note that higher rotation orders introduce greater computational costs. Consequently, existing models typically use a maximum rotation order of three. The experimental results shown in Figure 4 demonstrate that our PACE model exhibits reasonable stability and robustness to variations in hyperparameter choices.

## E  Examples

Considering $|\mathcal{N}_i| = 2$, $r_{i,j=1} = (x_1, y_1, z_1)$ and $r_{i,j=2} = (x_2, y_2, z_2)$, then

$$\hat{f}_{\mathbf{x}_i} = \sum_{j \in \mathcal{N}_i} Y^{\ell=1}(\hat{r}_{ij})^{\otimes 2} = \mathbf{C}_{l=1}\mathbf{C}_{l=1}^T \begin{bmatrix} y_1^2 + y_2^2 & y_1 z_1 + y_2 z_2 & y_1 x_1 + y_2 x_2 \\ z_1 y_1 + z_2 y_2 & z_1^2 + z_2^2 & z_1 x_1 + z_2 x_2 \\ z_1 y_1 + z_2 y_2 & x_1 z_1 + x_2 z_2 & x_1^2 + x_2^2 \end{bmatrix}, \quad (41)$$

$$f'_{\mathbf{x}_i} = \sum_{j_1 \in \mathcal{N}_i} Y^{\ell=1}(\hat{r}_{ij_1}) \otimes \sum_{j_2 \in \mathcal{N}_i} Y^{\ell=1}(\hat{r}_{ij_2}) = \mathbf{C}_{l=1}\mathbf{C}_{l=1}^T \cdot \begin{bmatrix} y_1 + y_2 \\ z_1 + z_2 \\ x_1 + x_2 \end{bmatrix} \otimes \begin{bmatrix} y_1 + y_2 \\ z_1 + z_2 \\ x_1 + x_2 \end{bmatrix}. \quad (42)$$

For $\hat{f}_{\mathbf{x}_i}$, it has element $y_1^2 + y_2^2$, while $f'_{\mathbf{x}_i}$ only contains elements like $y_1^2 + y_2^2 + 2y_1 y_2$. Thus, $\hat{f}_{\mathbf{x}_i}$ can not be represented as a linear combination of elements in $f'_{\mathbf{x}_i}$.

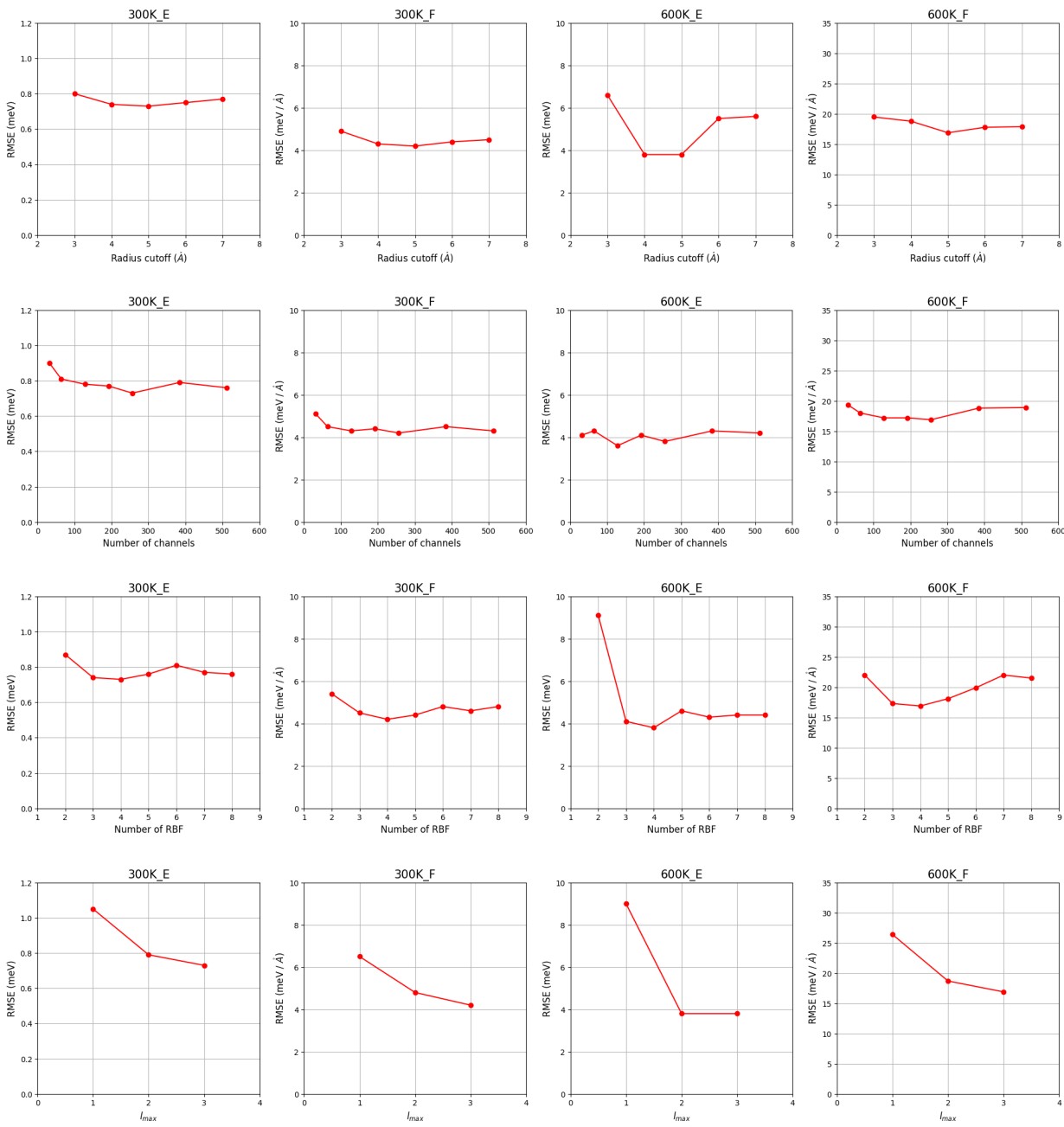

Figure 4: Exploration of hyperparameter space.

