# OpenReview forum: "Equivariant Graph Network Approximations of High-Degree Polynomials for Force Field Prediction"
_TMLR — Accepted by TMLR_

### Review · Reviewer_QPwE · 2024-05-29

**Summary Of Contributions:**

This paper proposes a novel equivariant network named PACE to better predict atomic potentials and force fields. PACE is based on polynomial function approximation and Atomic Cluster Expansion (ACE). First, it incorporates edge booster in the first layer to build edge messages from neighboring nodes. Next, the additional self-interaction (SI)layers are introduced in the polynomial interaction module. Empirical results validate the network's efficacy, demonstrating its robust performance.

**Audience:**

Yes

**Claims And Evidence:**

Yes

**Requested Changes:**

Please refer to *Weaknesses*.

**Strengths And Weaknesses:**

## Strengths

1. This paper is well-written with clear logic plus theoretical and empirical analysis.
2. The authors innovatively utilize Eq. (12) to (16) to construct their edge booster, which considers only spherical harmonics and atomic types without involving node features. Such design cleverly increases reusability and reduces computational costs.
3. The inclusion of additional self-interaction layers deepens the network's symmetry comprehension and expands its capacity to approximate a broader array of positions in D-spanning functions. This innovation is both novel and practical.
4. The authors present thorough empirical studies, the results of which are clearly SOTA.

## Weaknesses

This work is generally neatly presented. I only have several minor suggestions.

1. The authors discuss computational cost reductions in Sec.3.5. I recommend providing quantitative details on these reductions and any associated efficiency gains.
2. Chap.4 related works could be positioned before the introduction of the PACE architecture. This change would help readers better appreciate the advancements made with PACE.
3. I am confused by the color scheme in Fig.1. For example, the "Self-interaction" block is in dark blue in part A and B while light purple in part D. However, the block "Many-body interaction" is in the same light purple color in part B. Please consider a unified color scheme.

---

> ### Author Response · Authors · 2024-06-14
> **Response by Authors**
>
> Dear reviewer QPwE, thank you for your valuable comments! We have uploaded the revised paper, and below are our responses to your concerns and questions.
>
> **Q1.**
> Thanks for your advice. We have added a quantitative analysis of the computational cost incurred by our edge booster in Appendix D.2.3 and Table 8. This analysis of runtime and memory consumption is conducted using the AcAc dataset. Results are reported in seconds per epoch and megabytes (MB). A batch size of 5 is used, and the average time is calculated over 10 epochs, with validation occurring once every 2 epochs. Our PACE model comprises 2 layers. In the first experiment, we remove the EB module and use the original spherical harmonics for tensor products in both PACE layers. The second experiment employs the EB module only in the first layer, which corresponds to the architecture of our PACE. In the third experiment, we add an additional EB module in the second layer. Here, the second-layer EB module uses the updated node features output by the first layer, instead of the node embeddings based on atomic types. The results demonstrate that the EB module with two consecutive tensor product operations is computationally expensive. Compared to incorporating the EB module into both layers, the design of using only one EB module in the first layer, and without considering updated node features, can efficiently reduce the time and memory cost by 30.5\% and 9.7\%, respectively.
>
> |               | No EB | EB in first layer | EB in both layers |
> |---------------|-------|-------------------|-------------------|
> | Training time | 12.1  | 17.8              | 25.6              |
> | Memory        | 2080  | 3076              | 3408              |
>
> **Q2.**
> Thanks for your advice. We have relocated the related works section (now Section 3) to precede the PACE architecture section (now Section 4).
>
> **Q3.**
> Thank you for pointing out the confusion caused by the color scheme! We have modified Figure 1 to ensure that each module or operation has a consistent color with its background. We use blue for the first layer (part A) and green for the second layer (part B). In the first layer, dark blue is used in the edge booster for distinguishing two tensor products. Since “Many-body interaction” is used in both layers, we use another color purple for its detailed zoom-out architecture illustrated in part D.
>
> We greatly appreciate the time and effort you've invested in reviewing our paper! We hope our responses have properly addressed your concerns. Should you have any additional comments or require further revision, we are readily available for further discussion and improvement. Thank you!
>
> Best regards,
>
> Authors

---

### Review · Reviewer_dhJf · 2024-05-31

**Summary Of Contributions:**

Summary:
This paper studies the equivariant graph network approximation, which largely relies on the effectiveness of spherical harmonics (SH) and tensor products (TP). This paper studies the equivariant polynomial functions for the equivariant architecture by proposing a novel framework called PACE which utilizes edge booster and Atomic Cluster Expansion (ACE) to approximate equivariant polynomial functions. By conducting extensive experimental analysis, the learning performance and generalization effectiveness of PACE is carefully validated on typical benchmark datasets in predicting atomic energy and force fields.

**Audience:**

Yes

**Broader Impact Concerns:**

No ethical concerns.

**Claims And Evidence:**

Yes

**Requested Changes:**

Please see the weakness part.

**Strengths And Weaknesses:**

Strengths:
- This paper is well-written and well-organized.
- The experimental results are good, which surpass the baselines by a large margin.

Weaknesses:
- Though using polynomial approximation and ACE is innovative, what makes PACE shows better performance than existing methods is still not clearly explained. The claims of achieving state-of-the-art performance are not strongly justified. It is suggested to provide clear intuition about why PACE differs from or improves upon previous works.
- The theoretical justifications provided for the use of polynomial approximations and spherical harmonics are not adequately rigorous. The paper could benefit from a more detailed and formal mathematical exposition. Additionally, some key concepts, such as the D-spanning function sets and their relevance to the model's expressiveness, are not fully explained, leaving gaps in the reader's understanding
- While the paper claims strong generalization capabilities in out-of-distribution settings, the evidence provided is insufficient. The experiments lack diversity in molecular complexity and size, which raises questions about the model's robustness and applicability to a wider range of molecular systems. More diverse and challenging test cases should be included to convincingly demonstrate the effectiveness of out-of-distribution generalization.
- The paper does not provide enough information to ensure reproducibility. Details on the implementation, such as specific software versions, data preprocessing steps, and code availability, are not adequately provided.

---

> ### Author Response · Authors · 2024-06-14
> **Response by Authors - Q1**
>
> Dear reviewer dhJf, thank you for your valuable comments! We have uploaded the revised paper, and below are our responses to your concerns and questions.
>
> **Q1.**
> Compared to previous models, one key improvement of our PACE model stems from its ability to approximate higher-degree equivariant functions. Each NequIP [3] layer has aggregated equivariant features similar to Equation 4. Thus, a NequIP layer's output irreducible representations (irreps) computed via spherical harmonics (Lemma B.1) and tensor product (Lemma B.2) can cover the equivariant polynomials defined as
>
> $Q^{(\mathbf{t})}\_{K=1} = \sum_j \mathbf{r}_{ij}^{\otimes t}, \quad t \leq \ell \_{\text{max}}$.
>
> However, such irreps cannot include some equivariant polynomials with the highest degree $D=2$, such as those illustrated in Equation 5. Consequently, the output of a NequIP layer can only form a $D$-spanning family with the highest degree $D=1$. Based on NequIP, the MACE [4] layer further includes a symmetric contraction module (Appendix B.2.2), which enables covering more equivariant polynomials represented as
>
> $Q^{(\mathbf{t})}\_{K=v\_{\text{max}}} = \sum\_{j\_1, \cdots, j\_{v\_{\text{max}}} \in \mathcal{N}\_i} \underbrace{\mathbf{r}\_{ij\_1}^{\otimes t\_1} \otimes \cdots \otimes \mathbf{r}\_{ij\_{v\_{\text{max}}}}^{\otimes t\_{v\_{\text{max}}}}}_{v\_{\text{max}} \text{ times}}, t\_1, \cdots, t\_{v\_{\text{max}}} \leq \ell\_{\text{max}}$.
>
> Hence, the output irreps of a MACE layer can form a $D$-spanning family with the highest degree $D=\min\\{\textcolor{red}{\ell\_{\text{max}}}, v\_{\text{max}}\\}$. Since further equipped with the edge booster, our PACE layer can cover equivariant polynomials with form
>
> $Q^{(\mathbf{t})}\_{K=v\_{\text{max}}} = \sum\_{j\_1, \cdots, j\_{v\_{\text{max}}} \in \mathcal{N}\_i} \underbrace{\mathbf{r}\_{ij\_1}^{\otimes t\_1} \otimes \cdots \otimes \mathbf{r}\_{ij\_{v\_{\text{max}}}}^{\otimes t\_{v\_{\text{max}}}}}_{v\_{\text{max}} \text{ times}}, t\_1, \cdots, t\_{v\_{\text{max}}} \leq \ell\_{\text{max}} * N\_{\text{boost}}$.
>
> Thus, the output irreps of a PACE layer can form a $D$-spanning family with the highest degree $D=\min\\{\textcolor{red}{\ell\_{\text{max}} * N\_{\text{boost}}}, v\_{\text{max}}\\}$. During our experiments, we set $\ell\_{\text{max}}=3$ and $v\_{\text{max}}=3$ following previous works [3, 4] for a fair comparison. Although both MACE and PACE can form a $D$-spanning family with $D=3$, our PACE model can still approximate a greater number of polynomial functions with $D > 3$ in this scenario.
>
> Compared to MACE, another enhancement of our PACE comes from the additional self-interactions (SI) used in the polynomial many-body interaction module. To approximate $n$ different positions in $D$-spanning function, MACE needs $v\_{\text{max}} \times n$ channels, while PACE with additional SI only needs $n$ channels. This is because the SI in the polynomial many-body interaction module enables every single channel in PACE to approximate one position in $D$-spanning functions. The corresponding proof is shown in Appendix B.2.2. We summarize this comparison of PACE and previous models in Section 3.4 of the revised paper.

---

> ### Author Response · Authors · 2024-06-14
> **Response by Authors - Q2**
>
> **Q2.**
> Thanks for your comments. We have revised proof of Lemma B.1 in Appendix B.2.1 to clarify the connection between polynomial approximations and spherical harmonics. In summary, for $n\leq 2$, we compare the dimension of function space for homogeneous polynomials and spherical harmonics to prove that any homogeneous polynomial on sphere can be a linear combination of spherical harmonics. For $n>2$, using Lemma 4.1 from [6], we can prove by recursion that homogeneous polynomials can be expressed as a linear combination of spherical harmonics with radial basis functions. We have revised this proof to demonstrate that spherical harmonics with radial basis functions can be linearly combined to form polynomials not only for $D<2$ but also for $D< \ell\_{\text{max}}$. We have revised the related content (Sec 2.4, Sec 3.4, Sec 4.5), but still show that PACE can cover a greater number of $SE(3) \times S_n$​ equivariant polynomial functions with a higher degree.
>
> When discussing the expressiveness of equivariant networks, equivariant polynomial functions are commonly used to analyze these networks [1, 2]. According to Lemma 1 from [1], any continuous $G$-equivariant function can be uniformly approximated on compact sets by $G$-equivariant polynomials. Consequently, the ability of a model to approximate equivariant functions increases with its capacity to cover more equivariant polynomials. In this context, a $D$-spanning family refers to a set of functions that span the polynomial function space up to degree $D$. This implies that any $S_n$permutation equivariant and rotation invariant polynomial function up to degree $D$ can be represented as a linear combination of elements within the $D$-spanning family. Therefore, networks that can encompass the $D$-spanning family with a higher $D$ exhibit greater expressiveness. In Appendix A.5, under the mathematical definition of $D$-spanning, we added this paragraph to elucidate the relationship between $D$-spanning function sets and the model's expressiveness. Additionally, we included a reference to Appendix A.5 under Equation 9, where the concept of $D$-spanning first appears in the main text.

---

> ### Author Response · Authors · 2024-06-14
> **Response by Authors - Q3**
>
> **Q3.**
> The experiment of our work focuses on the molecular dynamic (MD) simulation of a single molecular system, where a specific model is trained for each molecule. This kind of experimental setting is commonly used in existing machine learning methods for MD simulation due to the unique challenges and requirements of accurately simulating molecular behavior [3, 4, 5]. Each molecule has distinct structural and dynamic properties, with complex interactions that can vary significantly. Training a model for a single molecule allows it to capture these specific characteristics more accurately, including intramolecular forces and the effects of different conformations. Additionally, obtaining high-quality training data for many molecules is challenging. Existing MD datasets typically provide various 3D structures (conformations) of the same molecule, divided into training, validation, and test sets. In practical applications, focusing on specific molecules, such as drug candidates or unique materials, ensures tailored and accurate predictions for each molecule's unique requirements. Consequently, training one model per molecule is widely adopted in ML methods to achieve higher accuracy and reliability, essential for precise molecular dynamics studies.
>
> Under the setting where a single model is trained for each molecule, we follow our baseline methods [3, 4, 5] to conduct out-of-distribution (OOD) generalization tests based on temperature variations since the geometric distributions will have shifts. This is motivated by many real-world applications involving temperature fluctuations, such as drug interactions within the human body or material behavior under different environmental conditions. Therefore, evaluating a model's robustness and extrapolation capabilities under varying temperature conditions is crucial for predicting molecular behavior accurately. Unlike the rMD17 dataset, which provides more rigid molecules, the 3BPA and AcAc datasets focus on more flexible molecules. For instance, the 3BPA molecule is characterized by three freely rotating angles, which primarily induce structural changes at varying temperatures. In the case of 3BPA and AcAc, the training sets are sampled at 300K, while the test sets are sampled at different temperatures.
>
> During the rebuttal, we evaluated our model on an additional test set for 3BPA. This additional test set includes optimized conformations with dihedral slices, where two dihedral angles are fixed, and the third angle varies from 0 to 360 degrees. Consequently, this test set probes regions of the potential energy surface (PES) that are distant from the training set. The results are presented in the table below and have been updated in Table 2 of the revised paper. Root-mean-square errors (RMSE) are reported for both energy (E) and force (F) predictions, with meV and meV/$\mathring{\mathrm{A}}$ as units, respectively. Standard deviations are calculated over three runs with different seeds.
>
> |   | NequIP | Allegro    | MACE       | Ours       |
> |---|--------|------------|------------|------------|
> | E | 23.3   | 16.3 (1.5) | 7.9 (0.6)  | **7.6 (0.4)**  |
> | F | 23.1   | 20.0 (1.2) | 16.5 (1.7) | **16.0 (0.5)** |
>
> Thank you for pointing out the confusion caused by our strong claim regarding generalization capabilities in out-of-distribution (OOD) settings. We have revised our claim to be more precise, stating that our model demonstrates good generalization capabilities in OOD temperature conditions. Additionally, we have included more details (abstract, introduction, Sec 5.2) about the OOD setting in the revised paper to ensure greater clarity. Modified text is highlighted in red.

---

> ### Author Response · Authors · 2024-06-14
> **Response by Authors - Q4 & Reference**
>
> **Q4.**
> The software and hardware versions are provided in the first paragraph of Section 5 of the Experiment. Additional experimental details, including hyperparameters, are provided in Appendix C.2 (Appendix C in the initial version). Thanks for your advice. During the rebuttal, we added data preprocessing steps and pseudo code of our model implementation in Appendix C.1 and Appendix C.3, respectively. Upon acceptance, we will release the reproducible code through GitHub.
>
> **References:**
>
> [1]. Dym, Nadav, and Haggai Maron. "On the universality of rotation equivariant point cloud networks."
>
> [2]. Segol, Nimrod, and Yaron Lipman. "On universal equivariant set networks."
>
> [3]. Musaelian, Albert, et al. "Learning local equivariant representations for large-scale atomistic dynamics." Nature Communications 14.1 (2023): 579.
>
> [4]. Batatia, Ilyes, et al. "The design space of e (3)-equivariant atom-centered interatomic potentials."
>
> [5]. Batatia, Ilyes, et al. "MACE: Higher order equivariant message passing neural networks for fast and accurate force fields." Advances in Neural Information Processing Systems 35 (2022): 11423-11436.
>
> [6] Spherical harmonics and approximations on the unit sphere: an introduction
>
>
> We greatly appreciate the time and effort you've invested in reviewing our paper! We hope our responses have properly addressed your concerns. Should you have any additional questions or require further clarification, we are readily available for further discussion. Thank you!
>
> Best regards,
>
> Authors

---

### Review · Reviewer_h3gn · 2024-06-01

**Summary Of Contributions:**

This submission introduces a novel equivariant network called PACE, which aims to improve the prediction of atomic energy and force fields in molecular dynamics simulations by integrating edge booster and Atomic Cluster Expansion (ACE) techniques. PACE extends the analysis of equivariant polynomial functions, approximating high-degree SE(3) × Sn equivariant polynomial functions. It demonstrates state-of-the-art performance and generalization capabilities on datasets like rMD17, 3BPA, and AcAc. By leveraging spherical harmonics and tensor products, PACE effectively incorporates rotational and permutational equivariance, significantly enhancing the prediction accuracy of atomic potentials and force fields.

**Audience:**

No

**Broader Impact Concerns:**

Not involved.

**Claims And Evidence:**

No

**Requested Changes:**

1. The paper would benefit from the inclusion of an algorithm table that outlines the steps of the PACE network. This would help clarify the implementation details and make it easier for readers to understand and reproduce the methodology.

2. The authors need to discuss the proposed network's time complexity. Analyzing the computational complexity of PACE, not only empirical evaluations but also theoretical discussions, would provide valuable insights into its efficiency and practicality for large-scale applications.

3. The paper currently lacks a thorough exploration of the hyperparameter space. Investigating the sensitivity of the model to different hyperparameter settings could provide insights into the stability and robustness of the network.

**Strengths And Weaknesses:**

Strengths:

1. The introduction of the PACE network, which integrates edge booster and Atomic Cluster Expansion (ACE) techniques, represents a novel approach in the field of molecular dynamics simulations.

2. The paper provides an in-depth analysis of equivariant polynomial functions, transitioning from point cloud networks to equivariant networks, which adds valuable theoretical insights.

---

> ### Author Response · Authors · 2024-06-14
> **Response by Authors**
>
> Dear reviewer h3gn, thank you for your valuable comments! We have uploaded the revised paper, and below are our responses to your concerns and questions.
>
> **Q1.**
> Thanks for your advice. We have included the algorithm table in Appendix C.3 of the revised paper. In this subsection of the PACE pseudocode, we have also added text summarizing the model pipeline and the implementation of our key contributions.
>
> **Q2.**
> Thanks for your advice. We have provided the time complexity for several key components of existing equivariant model architectures in Table 6 of Appendix D.2.1 of the revised paper. In this table, $C$ is the number of channels, $L$ is the maximum rotation order of the equivariant features, and $v$ denotes the correlation order. Our PACE model requires three tensor products (TPs) to obtain the edge features and two polynomial many-body interaction modules on the aggregated node features. Each tensor product has a time complexity of $O(C L^6)$. Compared to the many-body interaction module used in MACE with a time complexity of $O(C^2L + C L^{4v+2})$, our polynomial many-body interaction module has a time complexity of $O(N C L^6 + E C L^{4v+2})$, where $N$ denotes the number of nodes and $E$ denotes the number of edges.
>
> | Model component                  |       Time complexity      |
> |----------------------------------|:--------------------------:|
> | Self-interaction                 |         $O(C^2 L)$         |
> | Tensor product                   |         $O(C L^6)$         |
> | Polynomial many-body interaction | $O(v C^2 L + C L^{4v +2})$ |
> | MACE many-body interaction       |   $O(C^2L + C L^{4v+2})$   |
>
> **Q3.**
> Thanks for your advice. In the revised paper, we have included the experimental results of hyperparameter exploration in Appendix D.4 and presented the performance curves for various hyperparameter choices in Figure 4. From top to bottom, the results correspond to four hyperparameters: radius cutoff, number of channels, number of radial basis functions, and maximum rotation order. Each data point represents the average performance over three runs with different random seeds. These experimental results demonstrate that our PACE model exhibits reasonable stability and robustness to variations in hyperparameter choices.
>
> We greatly appreciate the time and effort you've invested in reviewing our paper! We hope our responses have properly addressed your concerns. Should you have any additional comments or require further revision, we are readily available for further discussion and improvement. Thank you!
>
> Best regards,
>
> Authors

---

### Decision · Action_Editor_yzXZ · 2024-08-01

**Recommendation:** Accept as is

**Comment:**

This paper studies the equivariant graph network approximation, which largely relies on the effectiveness of spherical harmonics (SH) and tensor products (TP). This paper studies the equivariant polynomial functions for the equivariant architecture by proposing a novel framework called PACE which utilizes edge booster and Atomic Cluster Expansion (ACE) to approximate equivariant polynomial functions. By conducting extensive experimental analysis, the learning performance and generalization effectiveness of PACE is carefully validated on typical benchmark datasets in predicting atomic energy and force fields. This paper is well-written and well-organized. The experimental results are good, which surpass the baselines by a large margin. The authors address reviewers' concerns well in their rebuttal. Thus, I would like to recommend accept as is.

**Audience:**

Yes

**Claims And Evidence:**

Yes